# Lumina-mGPT: Illuminate Flexible Photorealistic Text-to-Image Generation with Multimodal Generative Pretraining

## Abstract

We present Lumina-mGPT, a family of multimodal autoregressive models capable of various vision and language tasks, particularly excelling in generating flexible photorealistic images from text descriptions. By initializing from *multimodal Generative PreTraining* (mGPT), Lumina-mGPT demonstrates that decoder-only Autoregressive (AR) model can achieve image generation performance comparable to modern diffusion model with high efficiency through *Flexible Progressive Supervised Finetuning* (FP-SFT). Equipped with our proposed *Unambiguous image Representation* (Uni-Rep), Lumina-mGPT can flexibly generate high-quality images of varying aspect ratios. Building on the strong image generation capabilities, we further explore *Ominiponent Supervised Finetuning* (Omni-SFT), an initial attempt to elevate Lumina-mGPT into a unified multi-modal generalist. The resulting model demonstrates versatile multimodal capabilities, including visual generation tasks like text-to-image/multiview generation and controllable generation, visual recognition tasks like segmentation and depth estimation, and vision-language tasks like multi-turn visual question answering, casting light on the rosy potential of this direction. We release all code and checkpoints, hoping to facilitate the progress toward building artificial general intelligence.

## 1 Introduction

Seminal models, including DALL-E 3 (Betker et al., 2023), Stable Diffusion 3 (Esser et al., 2024), and SoRA (Brooks et al., 2024), have demonstrated superior performance in photorealistic image and video generation using diffusion-based generative modeling over continuous latent image features. In contrast, autoregressive (AR) generative models, which rely on "next-token prediction," have revolutionized text generation with groundbreaking reasoning abilities, as exemplified by models like GPT-4 (Achiam et al., 2023) and Gemini (Team et al., 2023), over discrete token representation. However, AR-based generative modeling over vector-quantized image features still lags far behind diffusion-based counterparts in terms of photorealistic and controllable image generation.

Although previous autoregressive efforts, such as DALL-E (Ramesh et al., 2021), CogView (Ding et al., 2021), Parti (Yu et al., 2022), OFA (Wang et al., 2022), Unified-IO (Lu et al., 2022; 2024a), LlamaGen (Sun et al., 2024), and Chameleon (Team, 2024a), have extensively explored generative modeling over vector-quantized image features following the paradigm of large language models (LLMs), their results on text-to-image generation have either been unsatisfactory or produced high-quality samples limited to academic benchmarks like ImageNet (Deng et al., 2009). We summarize the limitations of previous approaches: (1) AR-based image generation models often rely on randomly-initialized transformers, resulting in poor image quality and slow convergence due to the lack of pretrained representations; (2) the verbose encoder-decoder architectures used complicate the framework and hinder scalability and generalization to other tasks; (3) these models are limited in decoding resolution and flexibility, often producing low-quality images with fixed resolutions; (4) they lack task extensibility, restricting their applicability to a narrow range of scenarios without integration with other tasks such as dense labeling and controllable image generation. More detailed discussion of these challenges can be found in Appendix C.

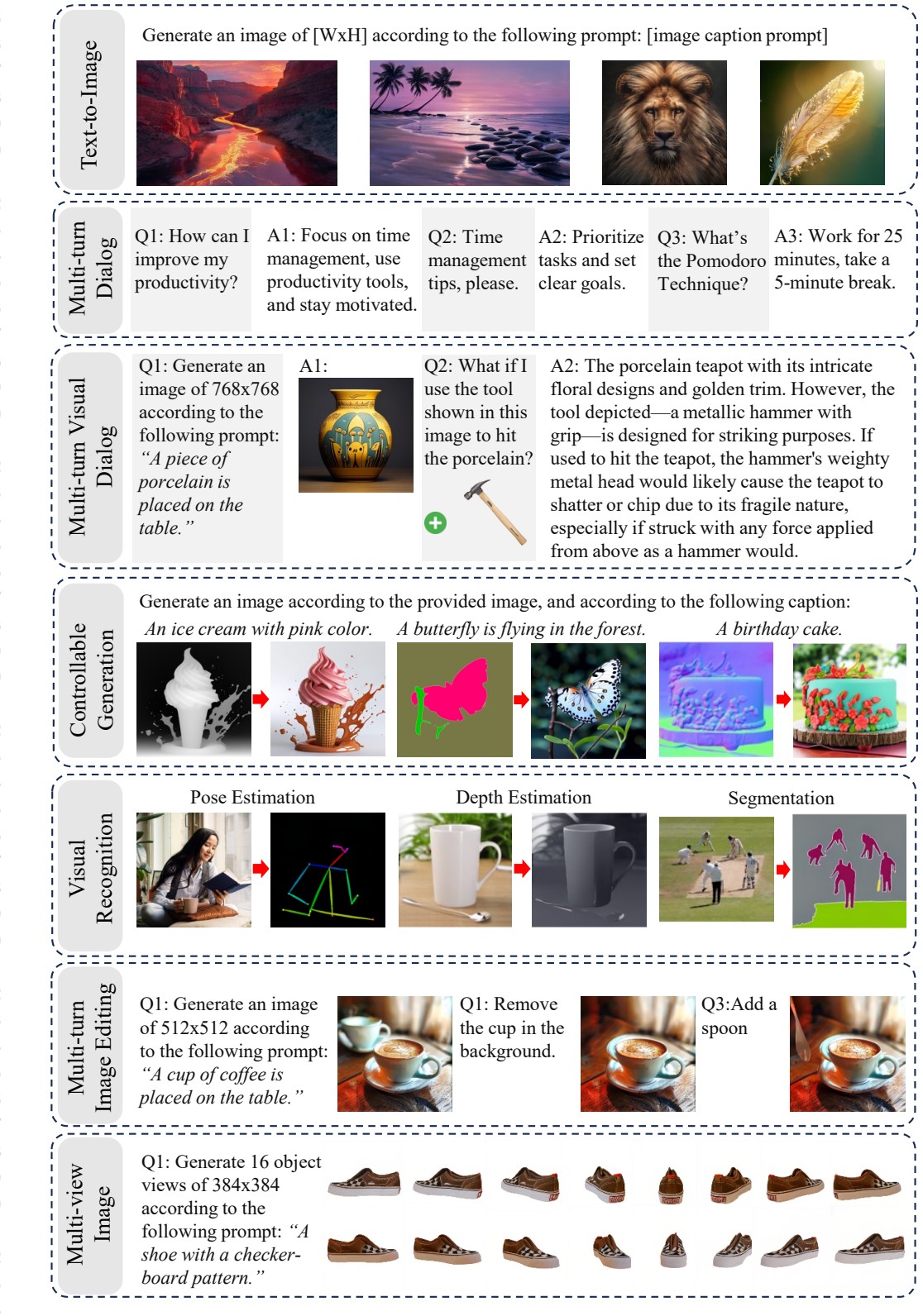

Figure 1: Unified in the next-token prediction framework, Lumina-mGPT can perform a wide range of multi-modal tasks. See Figure 13 to Figure 18 in the Appendix for more demonstrations.

Table 1: Overview of the design choices and capabilities of multimodal autoregressive approaches. Lumina-mGPT is the only model capable of both flexible photorealistic image generation and multimodal task unification, due to its decoder-only transformer design and multimodal generative pretraining.

| Method | Model Architecture | Multimodal Pretraining | Flexible Photorealistic Image Generation | Task Extensibility |
|---|---|---|---|---|
| DALL-E (Ramesh et al., 2021) | Decoder-only | ✗ | ✗ | ✗ |
| Cogview (Ding et al., 2021) | Decoder-only | ✗ | ✗ | ✗ |
| Parti (Yu et al., 2022) | Encoder-Decoder | ✗ | ✗ | ✗ |
| LlamaGen (Sun et al., 2024) | Encoder-Decoder | ✗ | ✗ | ✗ |
| OFA (Wang et al., 2022) | Encoder-Decoder | ✓ | ✗ | ✓ |
| Unified-IO (Lu et al., 2022) | Encoder-Decoder | ✗ | ✗ | ✓ |
| Unified-IO 2 (Lu et al., 2024a) | Encoder-Decoder | ✓ | ✗ | ✓ |
| CM3Leon (Yu et al., 2023) | Decoder-only | ✓ | ✗ | ✓ |
| Chameleon (Team, 2024a) | Decoder-only | ✓ | ✗ | ✗ |
| **Lumina-mGPT** | Decoder-only | ✓ | ✓ | ✓ |

To address the aforementioned challenges, we present Lumina-mGPT, a decoder-only transformer initiated with effective *multimodal **G**enerative **P**re**T**raining* (mGPT) and then supervised-finetuned over flexible, high-quality, high-resolution discrete image tokens in a progressive manner. This framework illuminates flexible high-resolution photorealistic image generation and can be easily extended to solve various component tasks in a unified manner. We provide a detailed comparison of the architecture design choices and model capabilities of existing multimodal autoregressive approaches in Table 1. The key features of Lumina-mGPT are outlined below:

① ***Effective Multimodal Generative Pretraining:*** Unlike commonly adopted approaches that utilize a randomly-initialized causal transformer to generate discrete image tokens in an autoregressive manner, Lumina-mGPT is initialized from an effective **m**ultimodal **G**enerative **P**ret**T**raining (mGPT) representation, which is a multimodal autoregressive transformer trained at large scale using a simple "next-token prediction" loss. We find that by leveraging the rich knowledge from large-scale multimodal pretraining, mGPT can learn broad and general multimodal representation. This serves as an excellent starting point, remarkably simplifying the task of text-to-image generation, accelerating training convergence on downstream tasks, and unifying various vision-language tasks. Due to the lack of training resources, this mGPT representation is directly adopted from the pretrained Chameleon 7B and 30B models released by Meta (Team, 2024a).

② ***Simple Decoder-Only Architecture:*** Lumina-mGPT adopts the simple decoder-only architecture. Compared to more complex designs like encoder-decoder architectures, decoder-only models have a significant innate advantage: they provide a simple, elegant, and extensible framework to unify various understanding and generation tasks across different modalities. This makes them a promising candidate for achieving true unification. Besides, by sharing the same architecture as the flourishing text-only LLMs, Lumina-mGPT can benefit from well-established theories and infrastructures in the LLM community, such as scaling properties (Chowdhery et al., 2022; Brown et al., 2020) and techniques to optimize the training and inference processes (Dao et al., 2022; Dao, 2024; Kwon et al., 2023). We train a family of Lumina-mGPTs with up to 30 billion parameters to provide a flexible trade-off between efficiency and performance and to explore the limit of its multimodal capabilities as model parameters scale.

③ ***Flexible High-quality Image Generation:*** Despite the aforementioned strengths of decoder-only architecture, the image generation capabilities of such models remain limited, creating a gap between the potential and the reality of this architecture. We thus propose Flexible Progressive Supervised Finetuning (FP-SFT) to fully fulfill the potential of high-quality text-to-image generation. This approach starts with low-resolution discrete tokens and progressively transitions to high-resolution discrete tokens. Combined with *Unambiguous image **Rep**resentation* (Uni-Rep), this weak-to-strong SFT strategy effectively grants the model with the ability to generate high-quality photorealistic images with flexible aspect ratios.

④ ***Omnipotent Task Unification:*** The high-quality image generation capabilities achieved during the FP-SFT stage provide the prerequisites for further exploring the fundamental advantages of the decoder-only architecture. We thus propose Omnipotent Supervised Finetuning (Omni-SFT), a

preliminary attempt to create an omnipotent generalist. Specifically, Omni-SFT treats various tasks–such as multi-turn dialog, visual multi-turn understanding, dense labeling, text-to-image generation, text-to-multiview generation, image editing, and spatial-conditional image generation–as a unified discrete modeling task, allowing the model to achieve omnipotent task unification via a natural language interface.

We summarize our contributions as follows: (1) We are the first (especially in the open-source domain) to demonstrate that a decoder-only AR model can achieve image generation performance comparable to modern diffusion models. Furthermore, by initializing from mGPT, the aforementioned capability can be achieved at low computational cost (32 A100 GPUs × 7 days for 7B model). (2) We propose UniRep, an image representation that empowers decoder-only AR models with the ability to flexibly generate images of varying aspect ratios. (3) Building on the strong image generation capabilities, we further explore Omni-SFT, an initial attempt to elevate the model into a unified generalist. Experiments underscore the promising potential of this direction. (4) We open-source the entire pipeline to encourage the community's further exploration of this topic.

## 2 METHODOLGY

Lumina-mGPT is a decoder-only transformer initialized with multimodal Generative PreTraining (mGPT) and finetuned over high-quality multimodal tokens derived from various tasks. Based on the robust mGPT representation and our proposed supervised finetuning strategies with unambiguous image representation, Lumina-mGPT achieves superior performance in photorealistic image generation and omnipotent task unification with high flexibility in image resolution and aspect ratio. In this section, we first introduce mGPT, followed by the training and inference details of Lumina-mGPT.

### 2.1 REVISITING MGPT WITH CHAMELEON

mGPT represents the family of models utilizing a decoder-only transformer architecture, pretrained on extensive multimodal token sequences. These models exhibit exceptional native multimodal understanding, generation, and reasoning capabilities, offering the potential for universal modeling across various modalities and tasks. We use the recent open-source model, Chameleon (Team, 2024a), as an example to illustrate the design choices and implementing details of mGPT.

**Multimodal Tokenization**  To unify text and images into a multimodal token sequence, it is essential first to tokenize both text and image into discrete space. Especially for images, the choice of tokenizer is crucial as it determines the upper limit of generation quality. Specifically, Chameleon trains a new byte pair encoding tokenizer for text. For images, it adopts the quantization-based tokenization method following prior works (Esser et al., 2021; Yu et al., 2022; Ramesh et al., 2021), converting continuous image patches into discrete tokens from a fixed codebook while reducing spatial dimensions. The quantized image tokens are then flattened into a 1D sequence and concatenated with text tokens in various ways to form a multimodal token sequence for unified modeling.

**Decoder-Only Transformer**  Unlike Unified-IO and Parti using an encoder-decoder architecture with pretrained encoders, mGPT trains a decoder-only autoregressive transformer from scratch after transforming text and image inputs into a unified sequence of discrete tokens $x = (x_1, x_2, ..., x_T)$, leading to a simpler and more unified approach for multimodal generative modeling. mGPT adopts a standard dense transformer architecture for scalability and generalizability, with minor adaptations like RoPE (Su et al., 2024) and SwiGLU (Shazeer, 2020) activation function, following the LLaMA series (Touvron et al., 2023). However, this standard transformer architecture exhibits training instability when scaling up the model size and context length of multimodal token sequences, as observed in existing works (Team, 2024a; Zhuo et al., 2024). It has been found that this instability is caused by the uncontrollable growth of network activations in transformer blocks. Therefore, Pre-Norm, Post-Norm, and QK-Norm (Henry et al., 2020) are added to each transformer block to preserve the magnitude of intermediate activation and further stabilize the training process.

**Training Objective**  During training, mGPT models the conditional probability $p(x_t|x_1, ..., x_{t-1})$ of multimodal sequences using the standard next-token prediction objective. Additionally, Chameleon applies z-loss (Chowdhery et al., 2022) to stabilize the training of 7B and 30B models. Initially, we

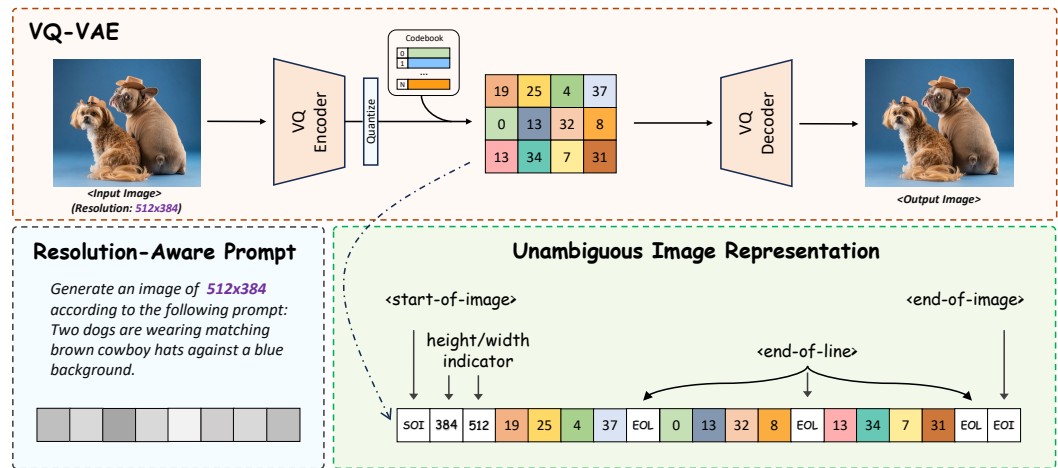

Figure 2: Illustration of Resolution-Aware Prompt (bottom left) and Unambiguous Image Representation (bottom right). These designs are used in all supervised finetuning stages to eliminate the ambiguity in image representation, enabling flexible resolution image modeling.

underestimated the importance of z-loss in our attempts because it is absent in most works relevant to the training of (multimodal) LLMs. However, we found that without this term, the magnitude of logits surges, causing the loss to diverge. On the other hand, with z-loss, we observe that for the 7B and 30B models, the optimal temperature for inference-time image generation is much lower, as the magnitude of logits is significantly reduced in the larger models.

**Limitations of Chameleon** Although Chameleon demonstrates potential for joint image and text understanding within one decoder-only transformer, its image generation ability remains inferior to state-of-the-art diffusion-based frameworks (Esser et al., 2024; Chen et al., 2024a; Li et al., 2024c; Team, 2024b; Zhuo et al., 2024) in both quality and resolution flexibility. Moreover, it is worth noting that the image generation ability is even absent in the open-source version of Chameleon. Additionally, the capabilities of Chameleon are confined to vision-language and text-only tasks, excluding a broader range of vision-centric tasks. These include classic visual recognition tasks such as segmentation and depth prediction, as well as creative visual generation tasks like controllable generation and image editing. Lumina-mGPT is built upon Chameleon to unlock its full potential for flexible photorealistic image generation and to become a versatile vision generalist.

## 2.2 LUMINA-MGPT

### 2.2.1 EFFECTIVE INITLIZATION

Large-scale pre-training and scalable model architecture have been widely verified as the golden path to advanced intelligence. As mGPTs like Chameleon are pretrained on large-scale image-text interleaved datasets and have developed effective and generalizable representations for both image and text, they can better serve as the starting point for flexible photorealistic image generation and beyond than random initialization or language-only models. Furthermore, the LLaMA (Touvron et al., 2023) architecture integrated with features like QK-Norm has demonstrated strength and scalability through extensive validation (Touvron et al., 2023; Meta, 2024; Sun et al., 2024; Gao et al., 2024). By initializing from the Chameleon mGPT, which adheres to the LLaMA architecture, we can leverage these architectural advantages. Consequently, initializing from mGPTs allowed us to efficiently train high-performing Lumina-mGPT models, with parameters ranging from 7B to 30B, using just 10M high-quality image-text data points.

### 2.2.2 SUPERVISED FINETUNING FOR LUMINA-MGPT

**Unambiguous Image Representation** Existing methods represent images as 1D flattened sequences of 2D discrete image codes. While adequate for fixed resolutions, this approach becomes

ambiguous when supporting variable resolutions, as with Lumina-mGPT. For instance, images with resolutions of $512 \times 512$, $256 \times 1024$, and $1024 \times 256$ can all be encoded into the same number of tokens, making it impossible to infer the original shape without examining the token contents. This ambiguity poses significant challenges for both image perception and generation.

To address this problem, we propose **Un**ambiguous **i**mage **Rep**resentation (Uni-Rep), which augments the image representations by adding extra height/width indicator tokens immediately after the `<start-of-image>` token and inserting `<end-of-line>` tokens after image tokens belonging to the same row. As shown in Figure 2, this modification ensures that the original shape of the images can be accurately parsed from the 1D representation without additional context or delving into the contents of the image tokens. This enhancement provides the foundation for Lumina-mGPT's ability to perform image-related tasks at any resolution and aspect ratio.

Note that while either the height/width indicators or the `<end-of-line>` tokens can independently achieve disambiguation, we still use both simultaneously because they have distinct benefits. When generating images, the height/width indicators, generated before any image tokens, pre-determine the shape of the image, aiding Lumina-mGPT in composing the image contents. On the other hand, the `<end-of-line>` tokens can serve as anchors, offering the 1D tokens sequence with additional explicit spatial information. We detailed in Section 3.3 to illustrate the role of these indicators.

**Flexible Progressive Supervised Finetuning (FP-SFT)**   The FP-SFT process equips the pretrained mGPT with the capability to generate high-resolution images with flexible aspect ratios in a progressive manner. The process is divided into three stages, where the product of width and height approximates $512^2$, $768^2$, and $1024^2$, respectively. In each stage, a set of candidate resolutions with similar areas but different height-width ratios are prepared, and each image is matched to the most suitable resolution. In the low-resolution stage, shorter sequence lengths and the resulting high training throughput allow the model to quickly traverse a large amount of data, learning the general composition of images and a broad spectrum of visual concepts. Conversely, in the high-resolution stage, the model is expected to focus on learning high-frequency fine-grained details unique to high-resolution images. Benefiting from the strong foundation built during the high-throughput pretraining and low-resolution finetuning stages, the low-throughput high-resolution finetuning stage is data-efficient, thereby enhancing the overall efficiency of the FP-SFT process.

A meticulously curated dataset of high-resolution photorealistic image-text pairs is used for FP-SFT. Moreover, the pure-text data from OpenHermess (Teknium, 2023) and the image-to-text data from Mini-Gemini (Li et al., 2024b) are also incorporated during training to prevent catastrophic forgetting. To provide users with a natural way to specify the desired resolution of generated images, we developed the resolution-aware prompt (Figure 2). For each image and its corresponding description, the prompt is structured as follows:

```
 Generate an image of {width}x{height} according to the following
prompt:  \n {description}
```

**Omnipotent Supervised Finetuning (Omni-SFT)**   While flexible photorealistic image generation is the primary target of Lumina-mGPT, we find that the resulting model after FP-SFT can be efficiently transferred to a wide spectrum of image understanding and generation tasks. We thus present Omni-SFT, a preliminary exploration toward boosting Lumina-mGPT to a visual generalist. Training tasks and data for Omni-SFT consists of the following:

1. Single- and multi-turn language-guided image-editing with data from MagicBrush (Zhang et al., 2024) and SEED (Ge et al., 2024) (only involving the real-world and multi-turn subsets).

2. Dense prediction tasks, including surface norm estimation from NYUv2 Silberman et al. (2012) and ScanNet Dai et al. (2017), depth estimation from Kitti v2 Cabon et al. (2020) and Sintel Butler et al. (2012), pose estimation from MSCOCO Lin et al. (2014), semantic segmentation data annotated with OneFormer (Jain et al., 2023) on image from Laion (Schuhmann et al., 2022), and grounding data from RefCOCO (Kazemzadeh et al., 2014).

3. In-house spatial-conditional image generation following ControlNet (Zhang et al., 2023), with conditions including surface norm, depth, pose, and segmentation.

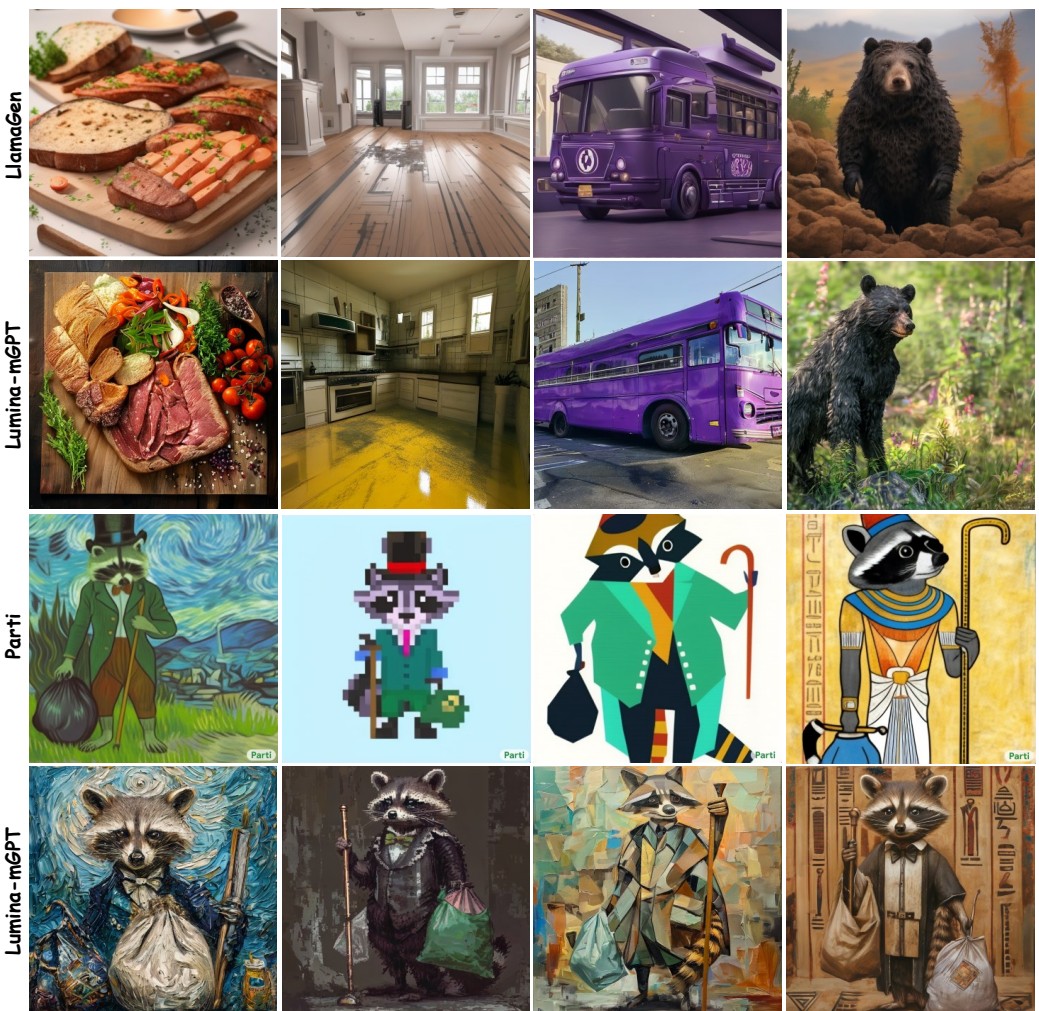

Figure 3: Qualitative comparison with LlamaGen and Parti. Lumina-mGPT can generate more photorealistic images with high aesthetics and fine-grained details.

4. Text-conditional multiview generation using an internal dataset consisting of 100k high-quality samples with rendered $384^2$ images from 16 viewpoints uniformly distributed in azimuth angles.

5. A small fraction of data sampled from those used in the previous FP-SFT process, including both text modeling and text-to-image generation to maintain its learned capabilities.

We tokenize all text and images into discrete tokens and formulate these tasks as a unified next-token prediction objective. Notably, we also incorporate tasks multiview generation, which requires generating a sequence of image frames, as a preliminary for video generation. As demonstrated in Section 3.2, after Omni-SFT, Lumina-mGPT exhibits a general capability for completing a wide range of tasks other than text-to-image generation, indicating the potential for building a multimodal generalist along this direction.

**Training Setup**    Though multiple tasks are involved in the SFT process, a unified next-token-prediction loss is used for all of the tasks. As Lumina-mGPT is designed as a chat model, all data are organized into single or multi-turn dialogs, with the loss applied only to the response parts. For all experiments, the AdamW (Loshchilov & Hutter, 2017) optimizer with weight decay = 0.1 and betas = $(0.9, 0.95)$ is used, and the learning rate is set to 2e-5. For stabilizing training, z-loss is applied with weight 1e-5 for both the 7B and the 30B model, and for the 7B model dropout with probability 0.05 is additionally applied. Inspired by the classifier-free guidance in diffusion models (Ho &

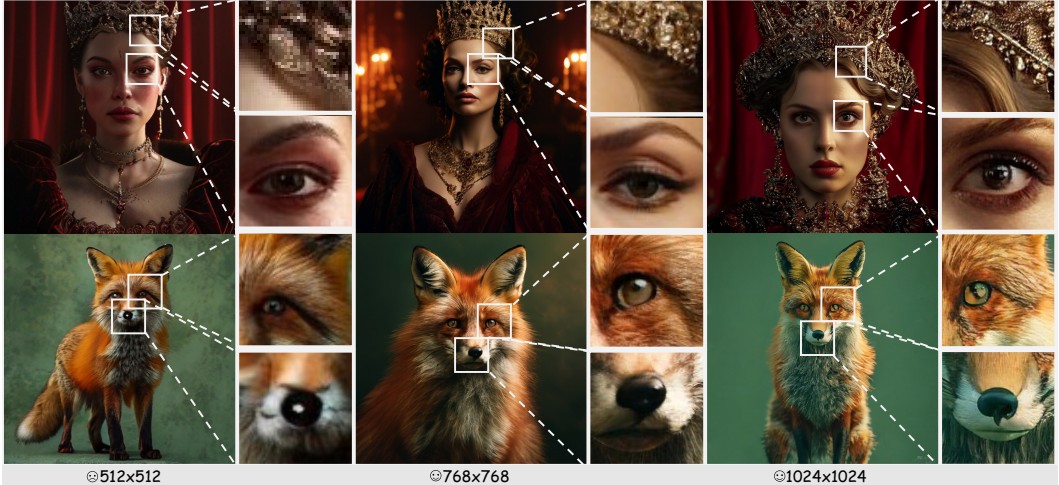

☺512x512 ☺768x768 ☺1024x1024

Figure 4: Samples with zoom-ins generated by Lumina-mGPT in different resolution finetuning stages. The visual details continuously improve along with the progressively increasing resolution.

Salimans, 2022), we randomly drop the context by a probability of 10% during training, as detailed in Appendix E. To accommodate the large model volume, PyTorch FSDP (Zhao et al., 2023) is employed with gradient checkpointing. To increase training throughput, all data are pre-tokenized before training and are clustered according to the number of tokens, ensuring that each global batch is composed of data with similar lengths.

## 3 EXPERIMENTS

### 3.1 FUNDAMENTAL PHOTOREALISTIC TEXT-TO-IMAGE GENERATION

We first demonstrate the fundamental text-to-image generation capabilities of Lumina-mGPT with FP-SFT. As shown in Figure 13, Lumina-mGPT can generate photorealistic images in a variety of resolutions, achieving the first native 1K autoregressive generation without cascaded models, a common technique in text-to-mage generation (Yu et al., 2022; Saharia et al., 2022; Chang et al., 2023; Pernias et al., 2024). These generated images exhibit strong semantic coherence with intricate visual details, despite being finetuned on limited computational resources and text-image pairs.

**Qualitative Comparison with SoTA AR-based Approaches** We compare the text-to-image synthesis ability of Lumina-mGPT with LlamaGen (Sun et al., 2024) and Parti (Yu et al., 2022). LlamaGen beats state-of-the-art diffusion models on ImageNet FID score. Compared with LlamaGen, Lumina-mGPT can achieve better visual quality as shown in Figure 3 in text-to-image generation. Note that Lumina-mGPT only requires 10M image-text pairs while LlamaGen is trained over 50M low-quality image-text pairs accompanied by 10M in-house aesthetic image-text pairs. Compared to Parti, an AR text-to-image model with 20 billion parameters, Lumina-mGPT also demonstrates better visual quality and aesthetics. However, due to significant differences in computational costs and training datasets, Lumina-mGPT demonstrates inferior text instruction following ability compared to Parti. In addition, neither LlamaGen nor Parti supports the end-to-end generation of 1K resolution images with arbitrary aspect ratios, as achieved by Lumina-mGPT. LlamaGen only supports a fixed resolution of $512 \times 512$, while Parti generates $1024 \times 1024$ images using an additional super-resolution upsampler. Beyond AR-based approaches, we also provide a side-by-side comparison with diffusion-based counterparts by training on the same dataset, as detailed in Appendix F.

**Quantitative Comparison on text-to-image benchmarks** We evaluate Lumina-mGPT on the popular text-to-image benchmarks T2I-CompBench (Huang et al., 2023), GenEval (Ghosh et al., 2023), and DPG-Bench (Hu et al., 2024) to objectively demonstrate its performance. Results are shown in Tab. 3.1. Lumina-mGPT shows clear improvement over Chameleon. Furthermore,

Lumina-mGPT outperforms Lumina-Next, a modern diffusion transformer trained using the same text-to-image data as Lumina-mGPT, and we guess that generative pre-training may be the reason behind this performance gap. Additionally, Lumina-mGPT shows competitive performance compared to SDXL, while maintaining a stable advantage over SDv2.1, providing an intuitive picture of Lumina-mGPT's position in the field.

Table 2: Quatitative results of Lumina-mGPT-7B on text-to-image benchmarks.

| | T2I-CompBench | | | GenEval | DPG-Bench |
|---|---|---|---|---|---|
| | Color | Shape | Texture | Overall | Average |
| Lumina-Next (Zhuo et al., 2024) | 0.5088 | 0.3386 | 0.4239 | 0.46 | 75.66 |
| SDv2.1 (Rombach et al., 2022) | 0.5694 | 0.4495 | 0.4982 | 0.50 | - |
| SDXL (Podell et al., 2023) | 0.6369 | 0.5408 | 0.5637 | 0.55 | 74.65 |
| Chameleon (Team, 2024a) | - | - | - | 0.39 | - |
| Lumina-mGPT (ours) | 0.6371 | 0.4727 | 0.6034 | 0.56 | 79.68 |

**On the Effectiveness of FP-SFT** To further validate the effectiveness of FP-SFT, we visualize the images generated at different finetuning stages in Figure 4. With increasing image resolution, we observe a progressive decrease in visual artifacts introduced by VQ-VAE and the emergence of diverse fine-grained visual details. From these illustrations, we can conclude that our FP-SFT can unleash the potential of generating high-quality images from mGPT in a progressive manner.

**Decoding Configuration Matters** In Appendix E, we highlight that optimal hyperparameters differ between text and image generation, with a status-aware control mechanism switching settings based on the type of content being generated. We then explore how different inference configurations, such as temperature, top-k, and classifier-free guidance scale, affect the quality of generated images. For example, lower temperatures and top-k values often result in over-smoothed images, while higher values enhance detail but may introduce artifacts. Inspired by these observations, we leverage min-p sampling (Nguyen et al., 2024) for visual generation and find it works significantly better compared to standard top-k sampling by striking the balance between coherence and diversity.

Table 3: Quatitative results of Lumina-mGPT-7B on comprehensive VQA benchmarks.

| | MMBench | MME-p | MME-r | SEEDBench-I | MMMU-val | POPE |
|---|---|---|---|---|---|---|
| Chameleon | 19.80 | 153.10 | 49.60 | 30.50 | 22.40 | 19.40 |
| Lumina-mGPT | **32.20** | **976.85** | **290.36** | **50.93** | **27.11** | **70.43** |

## 3.2 OMNIPOTENT TASK UNIFICATION WITH LUMINA-MGPT

By applying Omni-SFT over FP-SFT, Lumina-mGPT demonstrates a multitude of capabilities, which can be categorized into text-only multi-turn dialog, visual multi-turn dialog, multi-turn image editing, dense labeling, spatial-conditional image synthesis, and multiview generation. To intuitively illustrate these capabilities, we qualitatively visualize how various types of downstream tasks can be seamlessly integrated into Lumina-mGPT from Figure 14 to Figure 18 in the Appendix.

First, Lumina-mGPT effectively handles general text-only tasks in LLMs, such as solving math problems, coding, and commonsense reasoning, thanks to the extensive pertaining in Chameleon and our multi-task finetuning to mitigate catastrophe forgetting. As shown in Figure 14, Lumina-mGPT correctly answers "Which is bigger? 9.9 or 9.11", a question that has confused almost all existing LLMs including GPT-4 (Achiam et al., 2023) and Gemini (Team et al., 2023).

As depicted in Figure 15, Lumina-mGPT is also capable of handling various vision-language tasks including image captioning, visual question answering, and general multi-turn multi-image dialog. Quantitative results on multimodal understanding benchmarks (Liu et al., 2023b; Yue et al., 2024; Li et al., 2023; 2024a; Fu et al., 2023) presented in Table 3 illustrate that Omni-SFT significantly enhances the visual perception capabilities over the original Chameleon model.

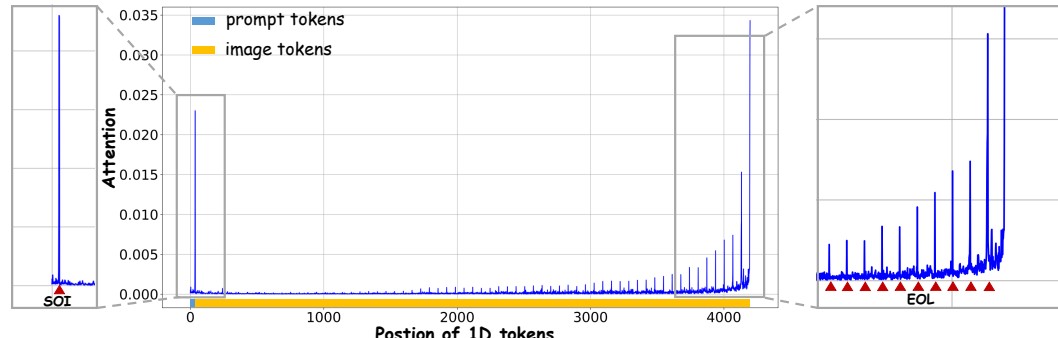

Figure 5: Visualization of averaged attention logits from the last image token. We observe "attention sinks" (Xiao et al., 2024), where added indicator tokens allocate a large proportion of attention score.

As a visual generalist, Lumina-mGPT incorporates classic visual recognition tasks. Using natural language as a unified interface, Lumina-mGPT can perform multiple high-level computer vision tasks including image segmentation, pose estimation, depth estimation, surface normal estimation, and referring object detection. See Figure 16 and 17 for examples.

For image generation, Lumina-mGPT supports both text and versatile spatial conditions, such as depth maps, segmentation maps, normal maps, and human poses, to guide the generation of target images, demonstrated in Figure 17 and 18. Beyond single-image generation, Lumina-mGPT can produce a sequence of consistent images from multiview, as shown in Figure 18. This task marks a preliminary step toward video generation by rendering the multiview images into a coherent video.

Given the above examples, though preliminary, they showcase that Lumina-mGPT can effectively follow diverse instructions, highlighting its promising potential as a unification of various challenging tasks in one framework.

## 3.3 ATTENTION VISUALIZATION

To better understand the sampling behavior of Lumina-mGPT, we visualize the average attention logits of the last image token during text-to-image generation, as shown in Figure 5. The results reveal that the attention score decreases for distant tokens, indicating that the model pays more attention to local tokens compared to distant ones. This behavior aligns with the long-term decay property designed in RoPE. Besides, we observe a similar pattern known as "attention sinks" in LLMs (Xiao et al., 2024), where a large proportion of attention score is allocated to a small number of tokens. After looking into details, these sink tokens are identified to be the indicators, including the `<start-of-image>` and `<end-of-line>` tokens. Notably, the attention scores of text tokens are significantly smaller than those of the `<start-of-image>` token. This suggests that most of the semantic information in text tokens may have been encapsulated into the `<start-of-image>` token. These intriguing findings indicate that Lumina-mGPT aggregates more information from these sink tokens, demonstrating the effectiveness of these indicators in our proposed Uni-Rep and opening up the potential for accelerating the sampling process with these sink tokens.

## 4 CONCOLUSION

In this work, we introduce Lumina-mGPT, a decoder-only transformer that can produce diverse, photo-realistic images at any resolution from text prompts. Instead of random initialization, Lumina-mGPT features initializing from an autoregressive transformer with multimodal Generative PreTraining (mGPT). Leveraging the general multimodal representation learned from massive interleaved data, we design two efficient finetuning strategies named FP-SFT and Omni-SFT to unleash the potential of mGPT on text-to-image generation and omnipotent task unification, respectively. We demonstrate Lumina-mGPT's broad multimodal capabilities across a wide range of tasks, showcasing its potential as a general vision-language assistant.

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

# A  RELATED WORK

**Multimodal Large Language Models**  Recent large language models (LLMs) (Chowdhery et al., 2022; Touvron et al., 2023; Brown et al., 2020; Achiam et al., 2023; Team et al., 2023) demonstrate strong instruction-following and reasoning capabilities, coupled with extensive world knowledge. To extend LLMs' expertise from the text domain to multimodal domains such as images and videos, prior works (Liu et al., 2023a; Lin et al., 2023a; Maaz et al., 2023; Lin et al., 2023b) have aligned pretrained encoders from various modalities with LLMs by curating multimodal instruction tuning datasets. Although these multimodal large language models (MLLMs) have exhibited powerful visual understanding capabilities, their multimodal functionality is primarily limited to perceiving the visual world, exemplified by tasks such as visual question answering and image captioning, rather than generating visual outputs. Another line of research Koh et al. (2024); Dong et al. (2024); Sun et al. (2023); Wu et al. has proposed augmenting MLLMs with the ability to generate images, videos, and audio from text instructions. These approaches introduce additional visual tokens for generation and align these generative tokens as conditional information with a pretrained generator, such as Stable Diffusion (Esser et al., 2024; Podell et al., 2023) for text-to-image generation. Consequently, the generation capabilities heavily rely on the external expert generator rather than MLLMs themselves, resulting in inconsistent and inferior generation results. To combine the strength of both approaches, our model aims to learn both understanding and generation of images using an MLLM with native multimodal capabilities, drawing inspiration from Chameleon (Team, 2024a), a mixed-modal early-fusion foundation model.

**Text-to-Image Generation**  The task of text-to-image generation seeks to synthesize photorealistic and diverse images based on textual descriptions. Nowadays, diffusion models, whether in pixel space (Saharia et al., 2022) or in latent space (Podell et al., 2023; Esser et al., 2024; Pernias et al., 2024), have become the de-facto approaches due to their superior performance, particularly in producing extremely high-aesthetic images. Among these models, the recent trend of scaling diffusion transformers (DiTs) (Chen et al., 2024b;a; Gao et al., 2024; Esser et al., 2024; Li et al., 2024c; Team, 2024b) suggests a unified architecture for both text and image modeling. However, existing DiTs still leverage separate language models, such as CLIP (Radford et al., 2021) or T5 (Raffel et al., 2020), as text encoders. This modality gap between text and image representations not only leads to inaccurate generation but also hinders the development of a unified multimodal foundational generative model. Compared to the dominance of diffusion models, the progress of autoregressive image generation has received less attention in recent years. Early works (Ramesh et al., 2021; Ding et al., 2021) proposed a two-stage generation approach: first, training a VQ-VAE Van Den Oord et al. (2017); Esser et al. (2021) for image tokenization and de-tokenization, and then using an autoregressive transformer to model discrete image token sequences, akin to language modeling. Parti (Yu et al., 2022) scaled up the autoregressive transformer to 20 billion parameters, demonstrating promising high-fidelity image generation results. LlamaGen (Sun et al., 2024) further improved the image tokenizer and integrated advanced techniques in LLMs, bridging the performance gap with diffusion counterparts. Unlike Parti and LlamaGen, Lumina-mGPT proposes multimodal generative pertaining on unified text-image sequences, followed by supervised finetuning on high-quality text-to-image pairs, achieving flexible high-aesthetic image generation with autoregressive models.

# B  DISCUSSIONS ON RECONSTRUCTION QUALITY

VQ-VAEs (Van Den Oord et al., 2017; Razavi et al., 2019; Esser et al., 2021) compress images at the cost of information loss, which introduces quality degradation at reconstruction, especially for high-frequency details such as edges, hair, and text. As Generative models such as Lumina-mGPT only has access to the VQ-VAE latents during training and cannot access original images, intuitively the VQ-VAE reconstruction quality should somehow build an upper bound for such models' image generation quality.

However, we observe an interesting and counter-intuitive phenomenon. Given two data flows:

1. Image $\xrightarrow{\text{VQVAE Encoder}}$ latent $\xrightarrow{\text{VQVAE Decoder}}$ Recon1

2. Image $\xrightarrow{\text{VQVAE Encoder}}$ latent $\xrightarrow{\text{Lumina-mGPT using editing system prompt with instruction "no edit"}}$ latent2 $\xrightarrow{\text{VQVAE Decoder}}$ Recon2

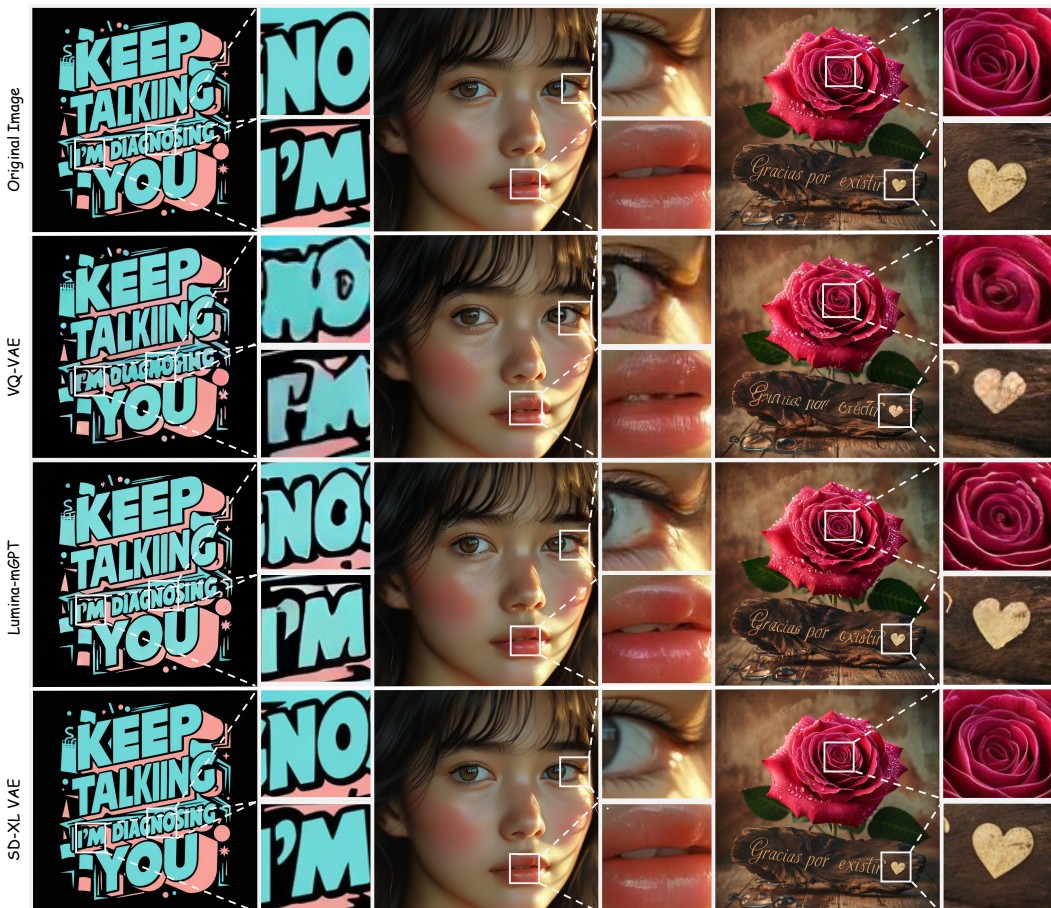

Figure 6: Reconstruction quality of different methods. Lumina-mGPT means first encode using VQ-VAE encoder, then send the latent to Lumina-mGPT using image editing system prompt with the instruction "no edit", and finally decode the newly generated latents using VQ-VAE decoder.

We surprisingly find that the quality of Recon2 sometimes surpasses that of Recon1, and we show such cases in Fig. 6. For reference, the reconstruction results using SDXL VAE (Podell et al., 2023) are also presented. Note that while Lumina-mGPT has been trained on the image editing task, it is not trained with the "no edit" instruction. This intriguing observation may suggest some meaningful insights. For example, it may possibly indicate that the latents encoded by VQ-VAE encoder could contain noises of certain patterns that can be learned and even corrected. It is worth noting that we believe this phenomenon should mainly be attributed to the intrinsic properties of VQ-VAE's discrete representation. We hypothesize that similar effects might also be observed in various generative models (e.g., autoregressive models, discrete diffusion models, etc.) and not due to any specific advantage or uniqueness of the decoder-only autoregressive architecture or other components of Lumina-mGPT, especially those claimed as contributions. We leave the further exploration of this phenomenon for future work.

## C LIMITATIONS OF EXISTING APPROACHES

**Randomly-Initialized Transformer** While transfer learning has revolutionized key fields such as visual recognition (He et al., 2016; Radford et al., 2021; Lu et al., 2019) and language generation (Raffel et al., 2020; Radford et al., 2018; 2019; Brown et al., 2020), popular autoregressive image generation approaches such as DALL-E, Parti, and LlamaGen all adopt a randomly-initialized causal transformer, which fails to utilize pretrained transferable representation and large-scale datasets. As a result, AR-based approaches often lead to poor image generation quality and slow convergence without leveraging proper large-scale pretraining.

**Verbose Encoder-Decoder Architeture** DALL-E and CogView initially propose using a decoder-only transformer for image generation with discrete representation, where a single transformer acts as both a text encoder and an image token decoder. However, subsequent approaches, such as Parti and LlamaGen, adopt a verbose

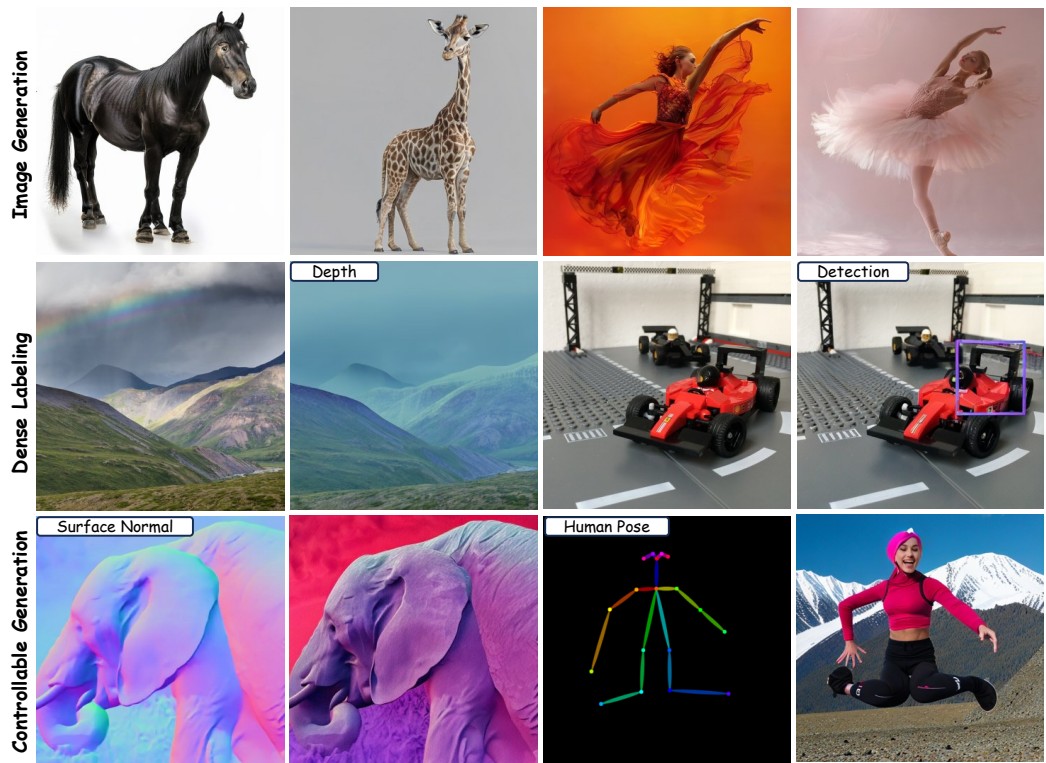

Figure 7: Failure cases of current Lumina-mGPT. Due to inadequate training and limited data size, Lumina-mGPT sometimes struggles to understand input conditions and produce visual artifacts.

encoder-decoder architecture that injects frozen T5 text features (Raffel et al., 2020) using cross-attention or prefix-filling approaches, motivated by the findings of Imagen (Saharia et al., 2022). Compared to the trend in scaling LLMs (Touvron et al., 2023; Bai et al., 2023; Bi et al., 2024), such encoder-decoder architecture is cumbersome due to the decoupling of text encoding and image token modeling. This design significantly complicates the autoregressive-decoding framework, limits the scalability of image generation, and hinders the generalization to additional modalities and tasks.

**Limited Decoding Resolution and Flexibility** Natural images exsit in various resolutions and aspect ratios. Advanced diffusion models (Chen et al., 2024a;b; Esser et al., 2024; Team, 2024b; Li et al., 2024c; Zhuo et al., 2024; Lu et al., 2024b) can successfully generate diverse photorealistic images at arbitrary resolution with skewed ratios. In contrast, current AR-based approaches (Yu et al., 2022; Team, 2024a; Sun et al., 2024) rely on central-cropping a low-resolution $512 \times 512$ image and transforming the cropped low-resolution image into a fixed-length sequence of discrete tokens using a pretrained Vector-Quantized Variational Autoencoder (VQ-VAE) (Van Den Oord et al., 2017; Razavi et al., 2019; Esser et al., 2021). This approach simplifies autoregressive training but at the cost of deteriorated image quality and generation flexibility.

**Poor Task Extensability** Autoregressive modeling excels at unified generative modeling for various tasks and modalities using discrete tokens (Wang et al., 2022; Lu et al., 2022; 2024a). However, previous AR-based image generation approaches (Ramesh et al., 2021; Ding et al., 2021; Yu et al., 2022; Sun et al., 2024) have been limited to text-to-image generation without exploring the unification with other tasks, such as dense labeling and controllable image generation. This lack of task extensibility significantly constrains the applicability of AR-based image generation models to a broader range of scenarios.

## D LIMITATIONS OF LUMINA-MGPT

**Failure Cases** Despite Lumina-mGPT can generate photorealistic images, it sometimes produces images with noticeable visual artifacts. For example, Lumina-mGPT may generate people and animals with unreasonable limbs, as shown in the first row of Figure 7. Besides, compared to SoTA text-to-image generation approaches including SD3 (Esser et al., 2024), Kolors (Team, 2024b), and HunyuanDiT (Li et al., 2024c), all pretrained over

1B image-text pairs, Lumina-mGPT's prompt-following ability is inferior due to the limited training resources and data size, which are many times smaller than these SoTA methods. Regarding dense labeling and controllable generation, Lumina-mGPT currently showcases preliminary results with a limited training budget. Hence, the second row in Figure 7 provides such an example where Lumina-mGPT produces inaccurate predictions or semantically inconsistent images, failing to understand the given image conditions. Therefore, we expect by scaling data size with more computational resources, Lumina-mGPT can effectively address the above failure cases such as inadequate instruction-following ability and visual artifacts.

**Generation Speed**    Autoregressive models require numerous network evaluations during inference due to the nature of next-token prediction, similar to the iterative denoising process in diffusion models. This becomes worse when generating high-resolution images, which often require minutes to generate a full sequence of image tokens, significantly slower than current diffusion models with advanced samplers. However, there have been plenty of techniques to optimize the inference speed designed for autoregressive models, such as vLLM (Kwon et al., 2023) and FlashAttention (Dao et al., 2022; Dao, 2024). We believe that by integrating these approaches in the future, Lumina-mGPT can achieve a remarkable speed up during inference.

**VQ-VAE Reconstruction Quality**    VQ-VAE is employed as the image tokenizer to convert continuous images into discrete token representations. Meanwhile, it also introduces information bottlenecks by compressing the spatial dimensions of images. As a result, the reconstruction quality of VQ-VAE largely determines the upper limit of generation quality. We discover that the VQ-VAE proposed in Chameleon sometimes struggles to reconstruct high-frequency details, especially when text and human faces are present in images. Incorporating further improvements on VQ-VAE, such as FSQ (Mentzer et al., 2023), may also enhance the generation quality of Lumina-mGPT.

# E    INFERENCE CONFIGURATION OF LUMINA-MGPT

In autoregressive models, various configuration parameters during Lumina-mGPT's decoding stage significantly affect sample quality (Holtzman et al., 2020; Radford et al., 2018; 2019). Hyperparameters such as temperature (T), top-k, and classifier-free guidance scale (CFG) have not been extensively investigated in the visual domain. In this section, we explore how these hyperparameters influence the generated image in terms of quality, texture, and style.

**Different Decoding Hyperparameters for Image and Text**    The sampling strategy of autoregressive models involves numerous hyperparameters that significantly influence the sampling results. We find that the optimal decoding hyperparameters differ greatly between text decoding and discrete image code decoding. For example, the top-k=5 setting performs well in generating text. However, when generating images, the value of top-k should be much larger (e.g. 2000) to avoid repetitive and meaningless patterns. Therefore, we implement a status-aware control mechanism for inference. Specifically, a set of default hyperparameters is used for text decoding; once a `<start-of-image>` token is generated, the hyperparameters switch to those optimized for image generation. After the `<end-of-image>` token is generated, the parameters revert to the initial settings.

**Classfier-Free Guidance**    Classifier-Free Guidance (CFG) (Ho & Salimans, 2022) is originally proposed to enhance the quality and text alignment of generated samples in text-to-image diffusion models. We incorporate this technique into autoregressive models during inference. When generating an image token, the CFG-processed logits $l_{\text{cfg}}$ are formulated as $l_{\text{cfg}} = l + s(l - l')$, where $l$ represents the original logits conditioned on the complete context; $l'$ represents the context-independent logits, which are conditioned solely on the tokens following the `<start-of-image>` token of the currently generating image, and are independent of any prior context; $s$ denotes the guidance scale of Classifier-Free Guidance. To make CFG work, during training, the context before <start-of-image> is randomly dropped by a probability of 10%. In practice, KV cache can be used for accelerating the computation of both $l$ and $l'$. As shown in Fig. 8, similar to the trend of diffusion models, increasing CFG initially raises the quality and stability of generation, but increasing it further would make the quality deteriorate.

**Influence of Temperature**    To evaluate the effect of these decoding parameters, we first set a standard decoding configuration: T=1.0, top-k=2000, CFG=4.0, which serves as a good-to-use setting. From this baseline, we gradually shift T from 0.7 to 1.0 to generate corresponding images at different temperatures. As shown in Figure 9, it is evident that when setting the temperature low, visual details diminish and objects tend to be over-smoothed. Conversely, when setting the temperature high, the generated images contain rich visual content but are prone to contain more artifacts.

**Influence of Top-k**    Based on the standard decoding setting, we vary the top-k value, from 50 to 8192, where 8192 is equal to the size of the VQ-VAE codebook usedin Lumina-mGPT. The results, visualized in Figure 9, indicate a similar trend with increasing temperature. When top-k is low, the image content and texture

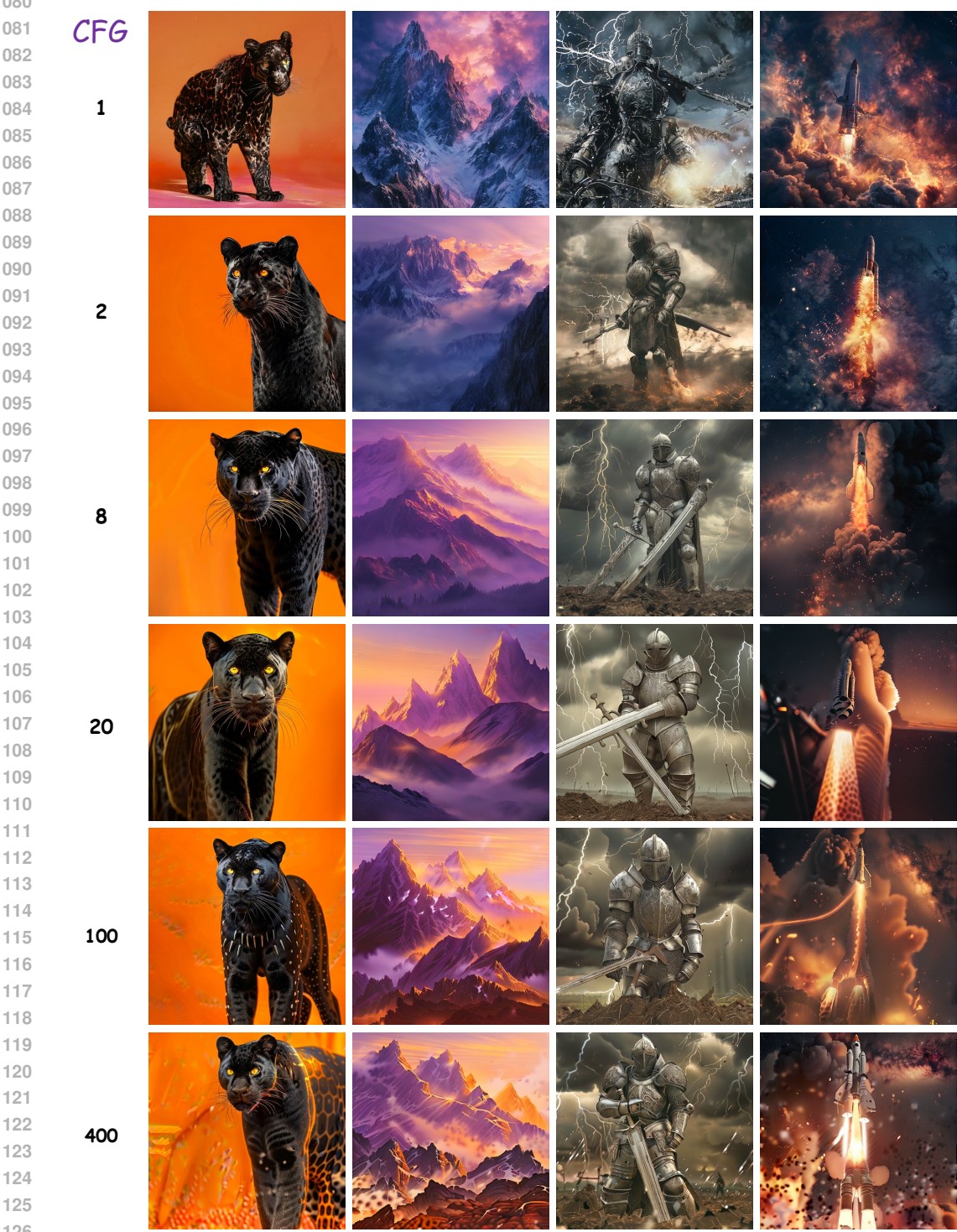

Figure 8: Samples generated by Lumina-mGPT using different CFG; T=1.0, Top-k=2000.

are relatively simple, exhibiting the over-smoothed problem as well. When top-k is set high, the image detail and texture are diverse, making it more aesthetically appealing, while increasing the potential of artifacts.

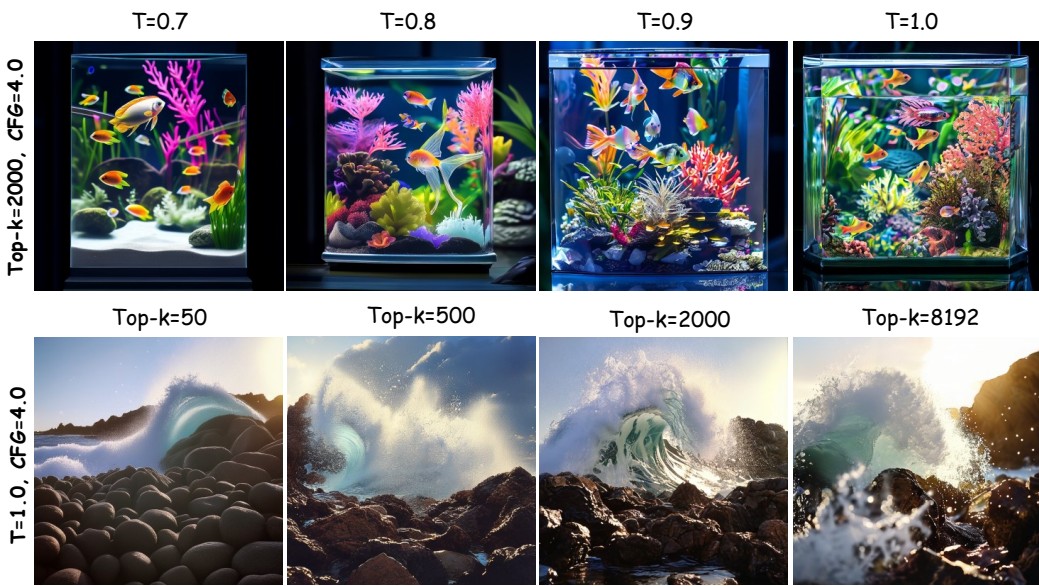

Figure 9: Samples generated by Lumina-mGPT using different Top-k and Temperature.

## F    COMPARISON WITH DIFFUSION-BASED APPRAOCHES

For a long period of time, diffusion models have dominated the field of text-to-image generation compared to autoregressive models. Although LlamaGen claims to beat diffusion models, their results are limited to the ImageNet benchmark and there has been no direct comparison between these two architectures so far. In this section, we aim to provide a detailed comparison of autoregressive and diffusion-based methods trained on these same text-image datasets, focusing on image quality, diversity, text-rendering, and multilingual capabilities. Specifically, we adopt Lumina-mGPT and Lumina-Next-SFT (Zhuo et al., 2024) as representatives of autoregressive and diffusion-based methods, respectively. A direct visual comparison between Lumina-Next-SFT and Lumina-mGPT reveals both the similarities and differences between autoregressive and diffusion-based generative modeling approaches.

**On the Similarity between Diffusion- and AR-based Generation** Given the same set of text prompts, both diffusion- and AR-based approaches generate photorealistic images with similar aesthetic style and fine-grained details, illustrated in Figure 10. This reveals the fact that both architectures can achieve satisfactory text-to-image generation performance when provided with the same training data, training budget, and comparable model sizes. The AR-based methods display remarkable visual aesthetics on par with their diffusion counterparts, challenging the notion that diffusion models are more effective and promising architecture in generative modeling. This finding also aligns with the platonic representation hypothesis (Huh et al., 2024) that neural networks are converging to learn a shared representation space despite being trained with different architectures and objectives. Therefore, this hypothesis highlights the importance of collecting more high-quality data and optimizing training infrastructure as directions for data and model scaling, to boost the overall model performance that is agnostic to any specific architecture.

**On the Differences between Diffusion- and AR-based Generation** As shown in Figure 10, Lumina-mGPT exhibits more diversity using different random seeds, while Lumina-Next-SFT generates similar images with identical layouts and textures. This can be partly attributed to the use of high temperature and top-k values in Lumina-mGPT. However, excessive diversity also causes our model to be less stable and more prone to producing visual artifacts, which is discussed in Section D.

We also compare the text rendering and multilingual understanding capabilities between Lumina-mGPT and Lumina-Next-SFT. As illustrated in Figure 11, Lumina-mGPT exhibits significantly better text synthesizing results, while Lumina-Next-SFT struggles to generate any complete character. We argue that this underscores the importance of mGPT, where the model learns a seamless multimodal representation between text and images using massive interleaved data during the pertaining stage. However, when it comes to multilingual understanding, Lumina-mGPT performs worse than Lumina-Next-SFT in terms of emoji and Chinese prompts shown in Figure 12. The reason is that although Lumina-mGPT learns better text-image alignment, the lack of multilingual text corpus used in pertaining limits its performance. In contrast, the text encoder used in Lumina-Next-SFT showcases significantly stronger multilingual capabilities than Chameleon. Hence, we hope

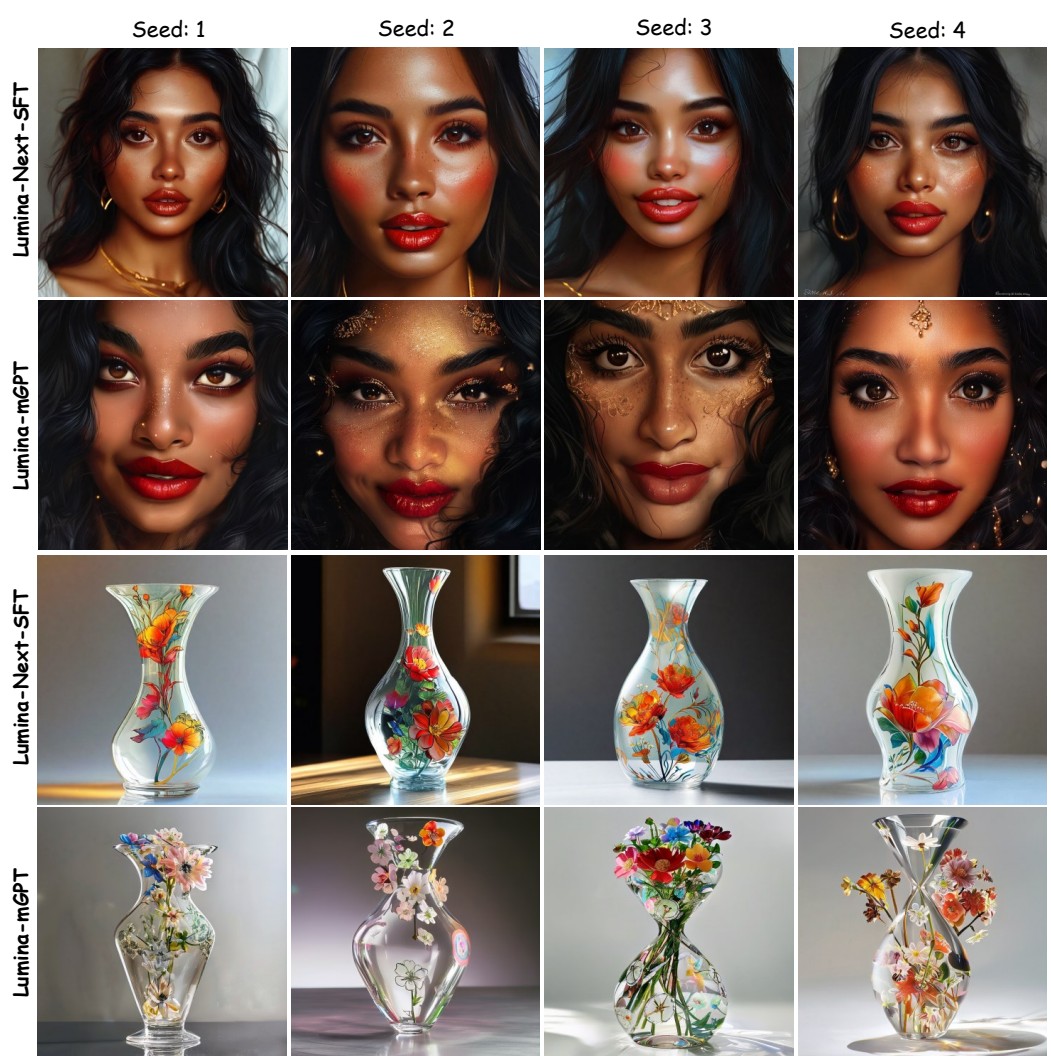

Figure 10: Comparison between Lumina-Next-SFT and Lumina-mGPT using different random seeds. Images generated by Lumina-mGPT exhibit comparable aesthetics with greater diversity.

that by comprehensively enhancing the capabilities of the base mGPT model, such as adding more multilingual data, Lumina-mGPT can benefit in all downstream tasks.

In addition to text-to-image generation, Lumina-mGPT supports various vision and language tasks within a unified framework. However, the design of diffusion models limits their compatibility and performance across multiple modalities and tasks. They often require specific architecture designs and additional training for each unseen task (Ke et al., 2024; Xu et al., 2023). In contrast, Lumina-mGPT treats input from all modalities as multimodal token sequences and leverages natural language as the interface to unify diverse tasks with next-token prediction.

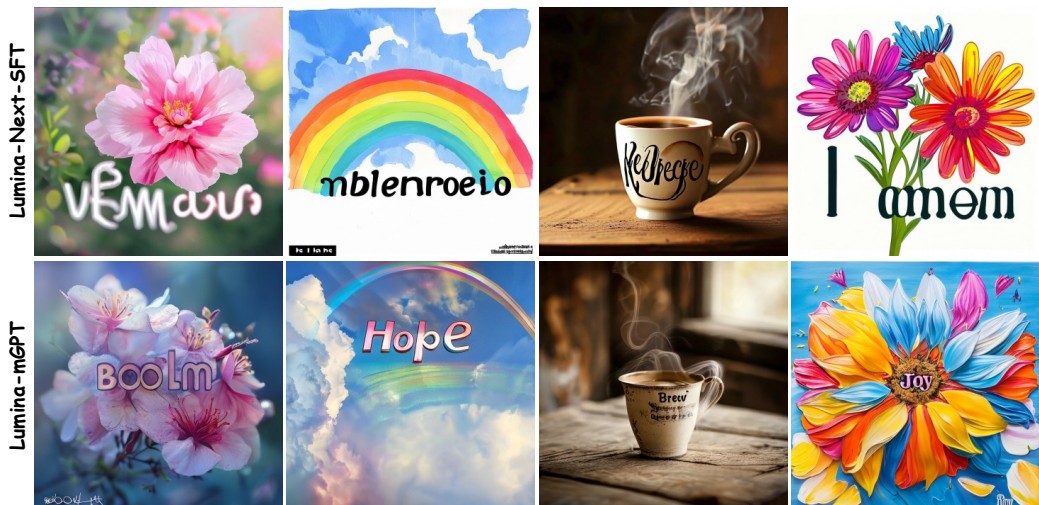

Figure 11: Text rendering comparison between Lumina-Next-SFT and Lumina-mGPT. From left to right, the correct texts to be rendered on the image are: "Bloom", "Hope", "Brew", and "Joy".

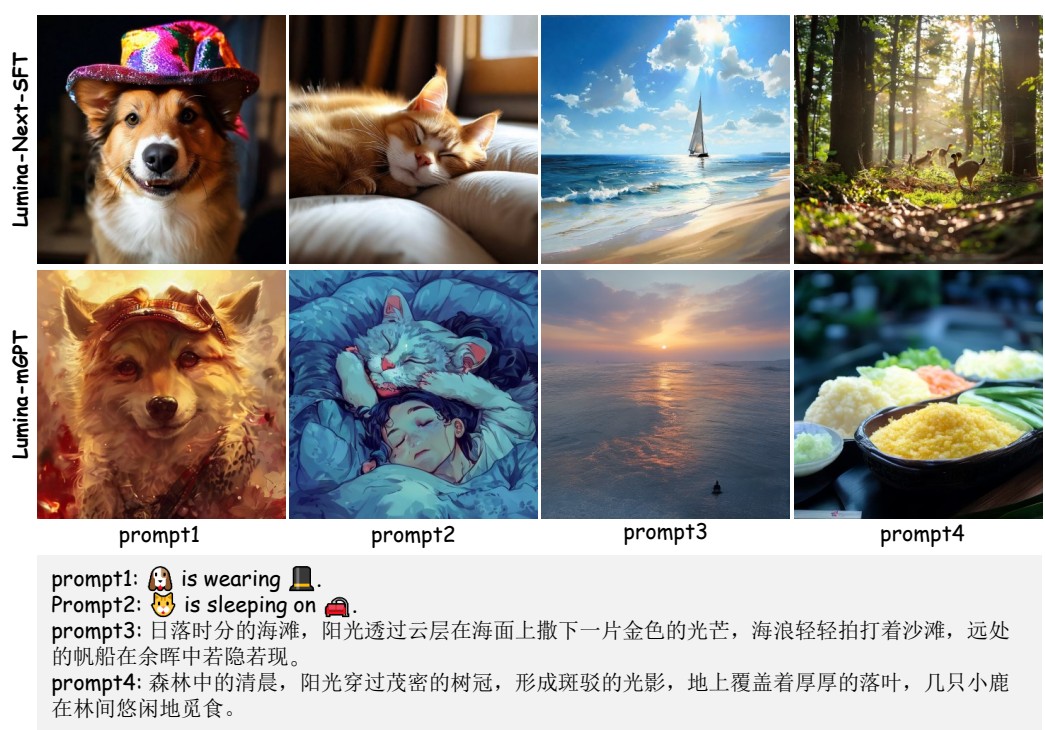

Figure 12: Emoji and multilingual instruction understanding ability comparison between Lumina-Next-SFT and Lumina-mGPT. Lumina-mGPT struggle to understand emojis and multilingual prompts.

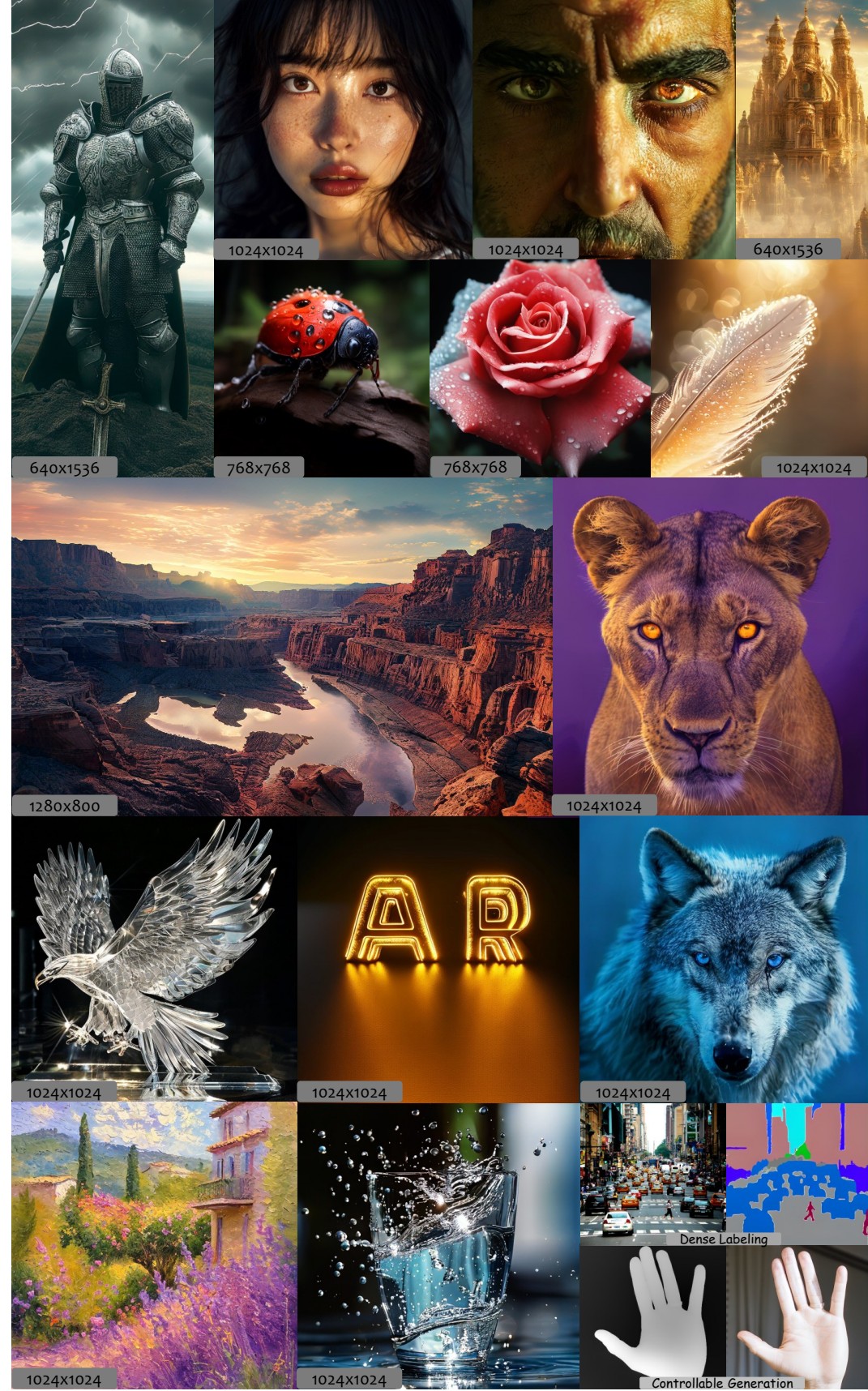

Figure 13: Selected photorealistic images of various resolutions generated by Lumina-mGPT. At the bottom, we include two examples to demonstrate the omnipotent task unification in Lumina-mGPT.

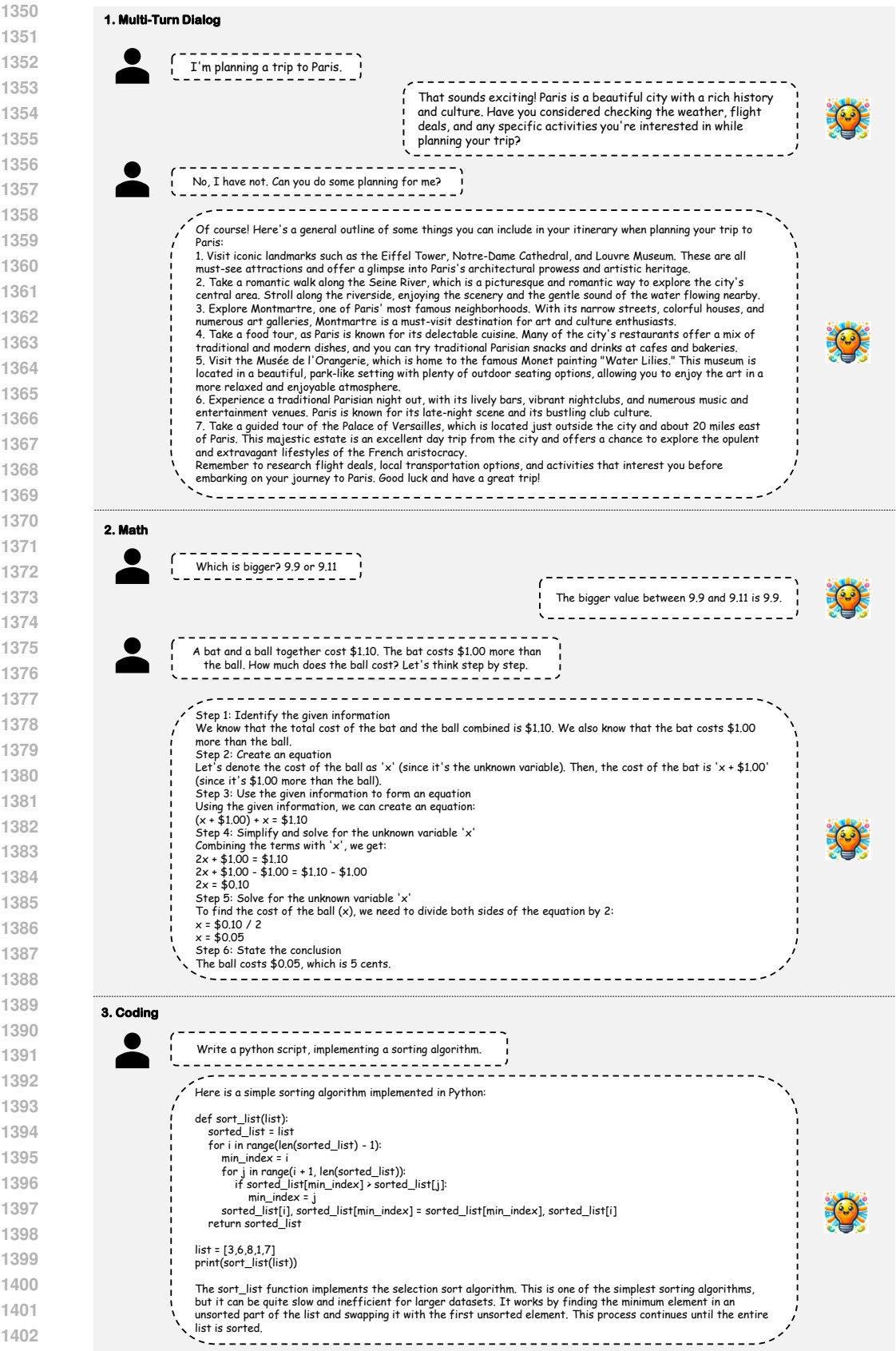

Figure 14: Lumina-mGPT as a generalist for various multimodal tasks.

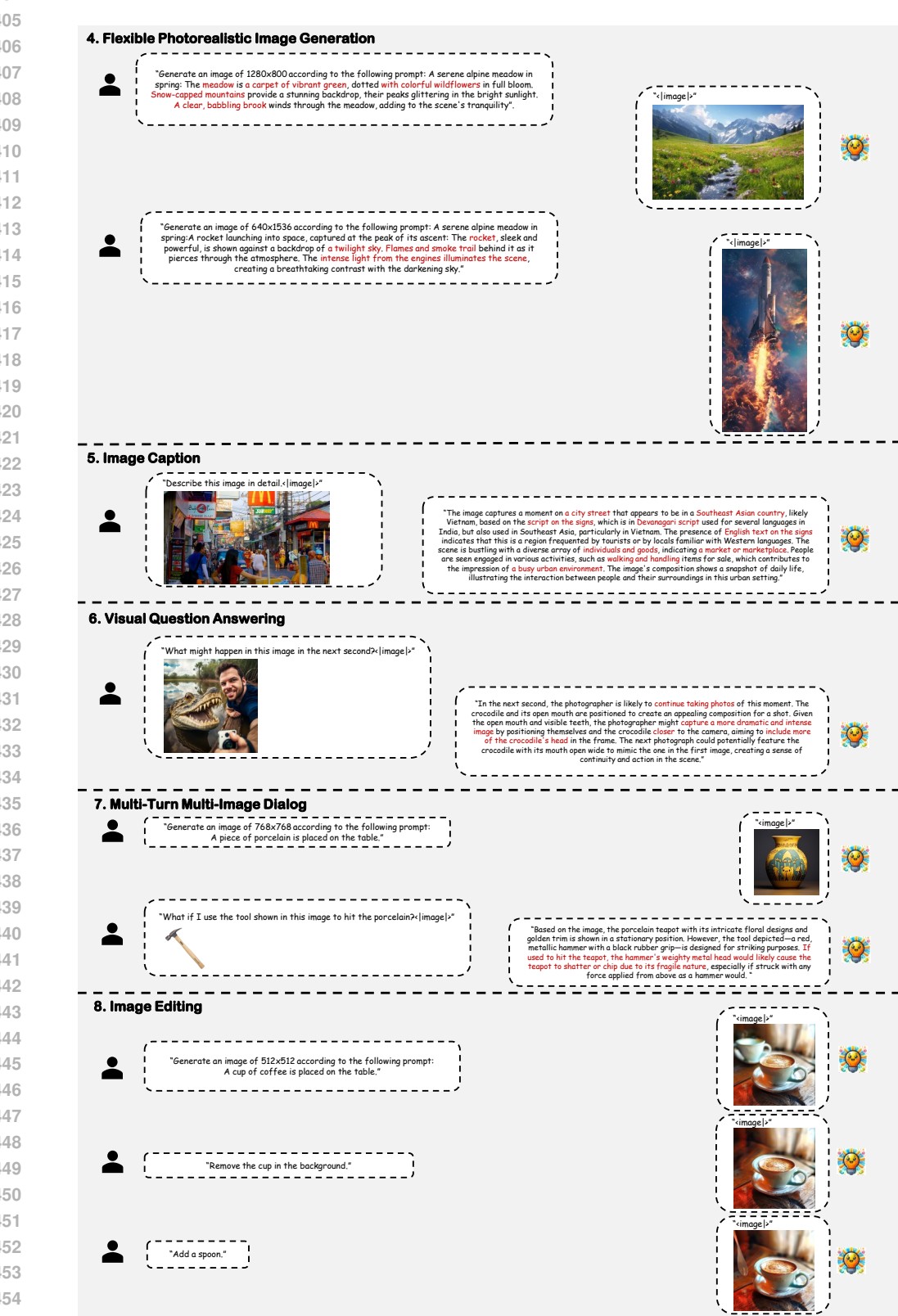

Figure 15: Lumina-mGPT as a generalist for various multimodal tasks.

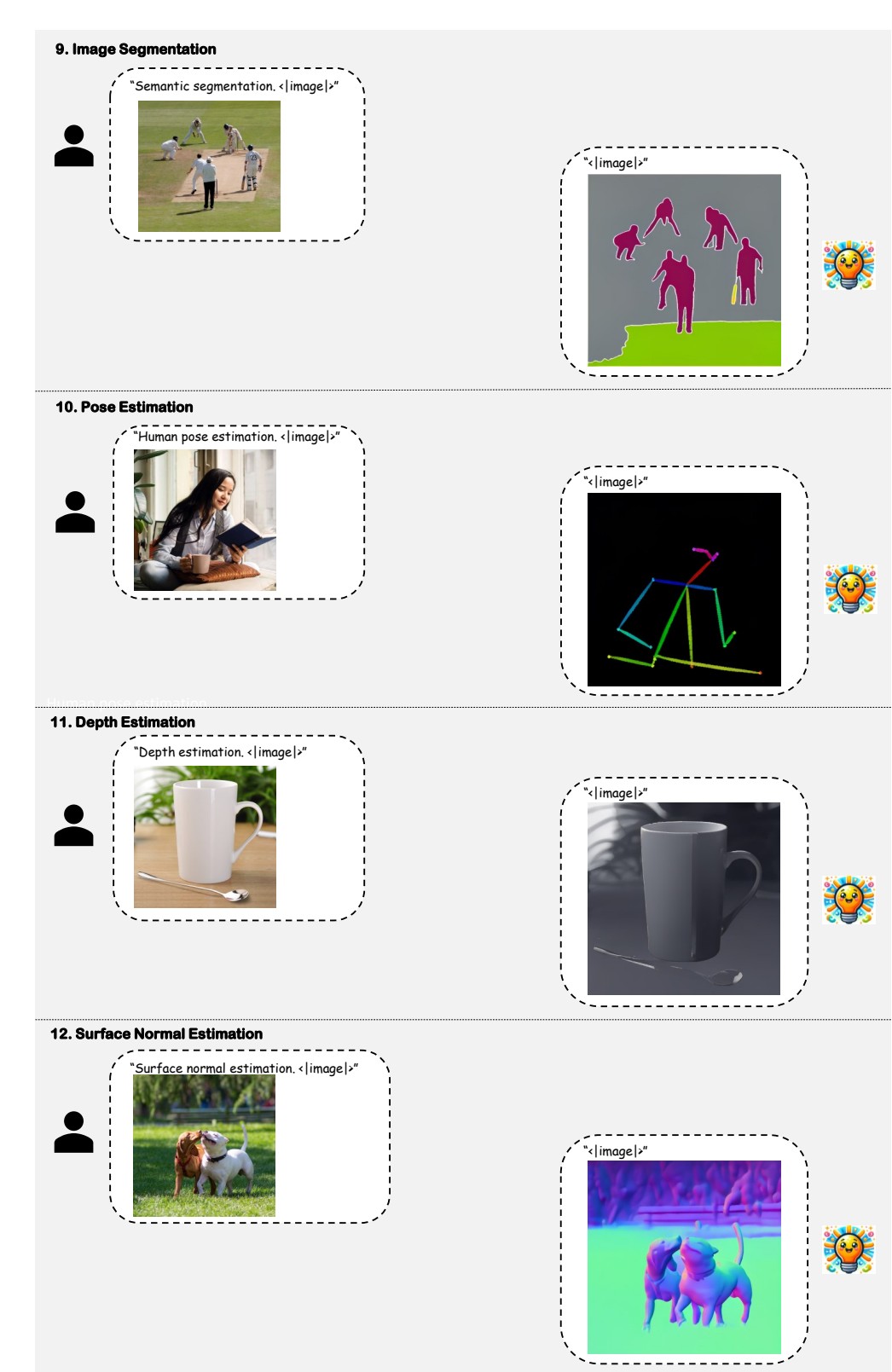

Figure 16: Lumina-mGPT as a generalist for various multimodal tasks.

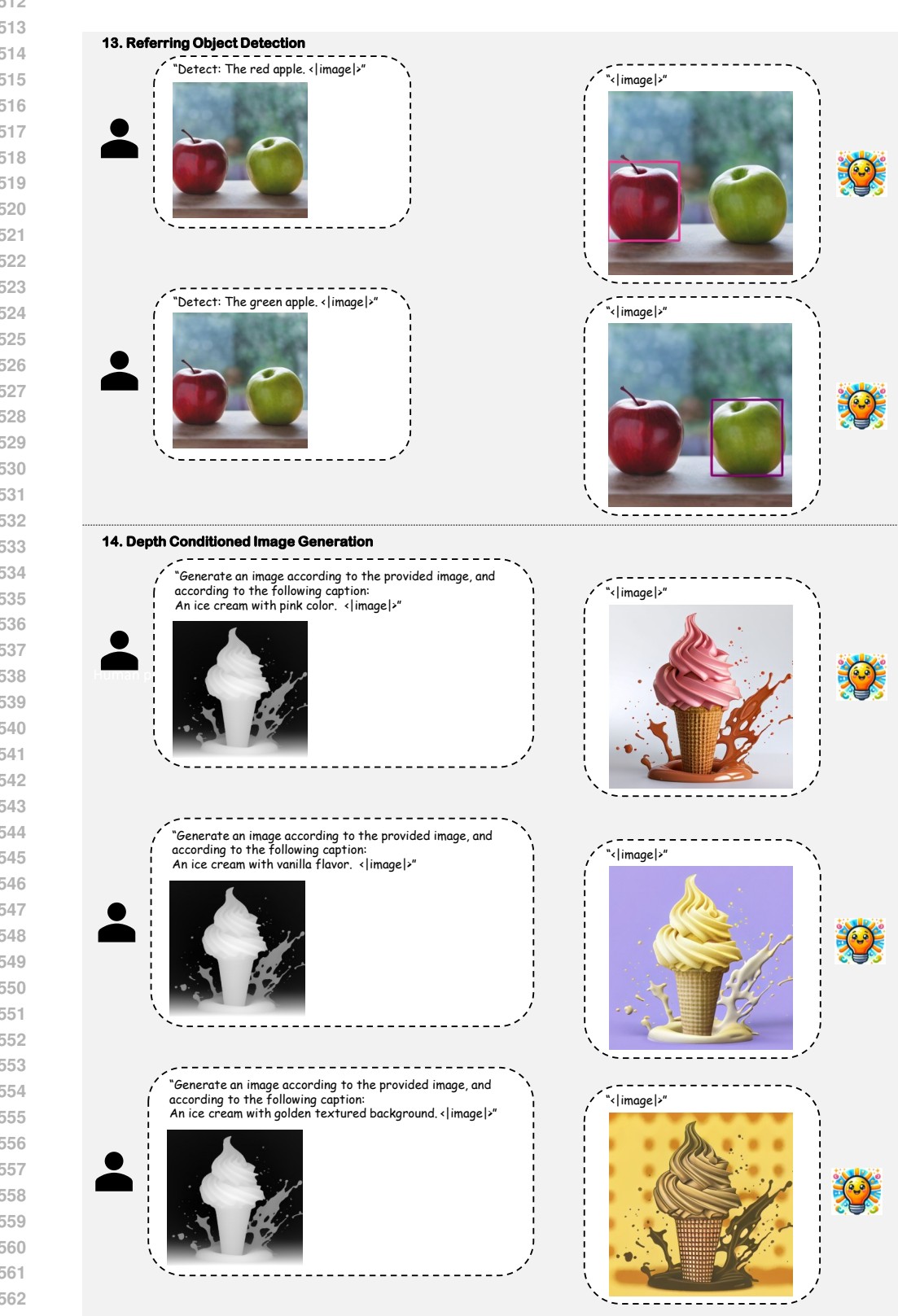

Figure 17: Lumina-mGPT as a generalist for various multimodal tasks.

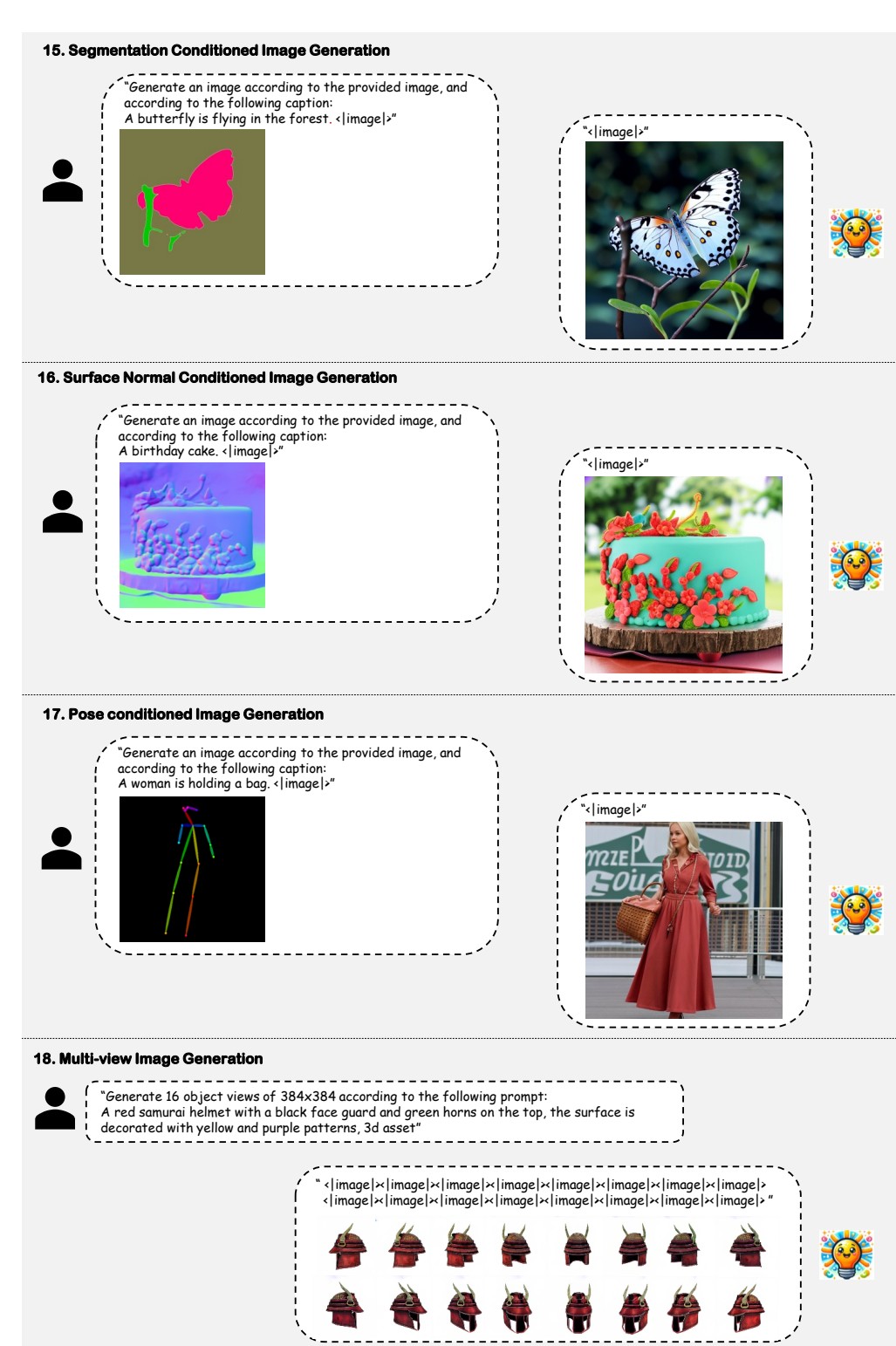

Figure 18: Lumina-mGPT as a generalist for various multimodal tasks.

