# OpenReview forum: "Lumina-mGPT: Illuminate Flexible Photorealistic Text-to-Image Generation with Multimodal Generative Pretraining"
_ICLR.cc/2025/Conference — Submitted to ICLR 2025_

### Official Review · Reviewer_c5qN · 2024-10-30

**Soundness:** 2
**Presentation:** 3
**Contribution:** 2
**Rating:** 5
**Confidence:** 4

**Summary:**

This paper introduces a large multimodal autoregressive model, Lumina-mGPT, with a particular focus on its text-to-image (T2I) generation capability after fine-tuning. The model leverages pre-trained token representations from Chameleon 7B and 30B (Meta) and incorporates two fine-tuning techniques—Flexible Progressive Supervised Fine-tuning (FP-SFT) and Omnipotent Supervised Fine-tuning (Omni-SFT)—to adapt the model for T2I tasks and multiple vision and language tasks.

Overall, I think this paper has both clear strengths and limitations, which I will detail in my comments below.

**Strengths:**

- The proposed model achieves very good image generation results on the T2I task in the AR paradigm, as stated as one of the main contributions.
- The paper presents a rather detailed discussion of the implementations and fine-tuning recipes, including failure cases, which is appreciated.
- This work is open-source, with the promise to release the model checkpoints and codes, thus, it may be of interest to the researchers in the open-source community on large models.

**Weaknesses:**

- While this may sound like a platitude, I must note that there is limited novelty in this work. As the authors acknowledge, a well-trained token representation for multimodal data is crucial in this setting, motivating their choice of Chameleon. The proposed fine-tuning techniques, however, follow commonly used methods and widely accepted design choices. Overall, the work reads very much like a technical report to me. However, I recognize that both intellectually novel approaches and engineering-heavy work have their place in the community. That being said, I am fine with either as long as the design choices are well-justified.
- A more concerning issue is the claim of the “emergent zero-shot capabilities.” While zero-shot is indeed becoming something of a buzzword, I don’t believe the observed reconstruction ability within the VQ-VAE module qualifies as true “zero-shot.” This term should be defined and examined rigorously, with careful attention to the training and inference data at different stages, especially within complex frameworks that combine multiple large models.
- While I appreciate the discussion on failure cases, Figure 6 actually highlights that this work does not address the fundamental challenges in generative modeling—specifically, the limitations in distribution learning with limited data. In other words, much of the performance boost appears to come from improved pre-trained data representations and high-quality, sufficient data during fine-tuning, rather than advancements in the model’s actual ability to learn distributions.

**Questions:**

Please see the questions from my review comments above. Overall, my preliminary rating is based on the pros and cons of the current work.

**Details Of Ethics Concerns:**

As this is an open-source large model for generation purposes, which may or may not be commercialized, I would suggest a further review of the potential legal and regulatory concerns.

---

> ### Author Response · Authors · 2024-11-20
> **Thank you for your comments!**
>
> ## About Novelty
>
> We acknowledge that our work is not driven by particularly complex or hard-to-conceive ideas. However, we firmly believe that it addresses previously unresolved problems and carries significant value for the field. We are delighted to see that you recognize the value of this type of work. For your information, given that several other reviewers pointed out that our initial submission lacked a clear summarization of contributions, we have clearly stated our contributions in the abstract and the introduction parts in the revision, and the relevant sections have been highlighted in red for easier review.  The summarized key contributions are as follows:
>
> 1. We are the first (especially in the open-source domain) to demonstrate that a decoder-only AR model can achieve image generation performance comparable to modern diffusion models. Furthermore, by initializing the model with multi-modal generative pretraining, this capability can be achieved with relatively low computational resources (32 A100 GPUs * 7 days for the complete 7B FP-SFT).
> 2. We propose UniRep, an image representation that empowers decoder-only AR models with the ability to flexibly generate images of varying aspect ratios.
> 3. Building on the strong image generation capabilities, we further explore Omni-SFT, an initial attempt to elevate the model into a unified, omni-potent model. Experimental results highlight the immense potential of decoder-only AR models in this direction.
> 4. We open-source the entire pipeline to encourage the community's further exploration of this topic.
>
>
>
> ## About the phrase *zero-shot*
>
> Thank you so much for pointing out this abuse of terms, and we totally agree with your opinion. We have carefully revised the expressions to avoid improper claims and potential misunderstandings, and the contents are now arranged in Appendix.B. In fact, by mentioning *zero-shot* in this manuscript, we referred to a specific phenomenon that we have observed:
>
>  given two data flows:
>
> 1. Image $\xrightarrow{\text{VQVAE Encoder}}$ latent$\xrightarrow{\text{VQVAE Decoder}}$ Reconstruction1
> 2. Image $\xrightarrow{\text{VQVAE Encoder}}$ latent$\xrightarrow{\text{Lumina-mGPT using editing system prompt with instruction "no-edit"}}$ latent2$\xrightarrow{\text{VQVAE Decoder}}$ Reconstruction2
>
> We surprisingly find that the quality of Reconstruction2 sometimes surpasses that of Reconstruction1. Note that, while we train on image editing tasks, the model is not trained on the "no-edit" instruction, which led us to originally use the term "zero-shot". This phenomenon is counter-intuitive, as Lumina-mGPT only has access to VQVAE latents during training and cannot access original image, so intuitively the quality of VQVAE reconstruction should somehow build an upper bound for Lumina-mGPT's image generation quality, which contradicts our observations. We find this observation intriguing and believe it may provide some insights (e.g. it may indicate that the latents encoded by VQVAE encoder could contain noises of certain pattern that can be learned and even correctly by Lumima-mGPT after training), so we added relevant contents to the manuscript. Overall, we believe that it would much more proper to claim it as *an interesting observation* instead of *a zero-shot ability*.
>
> ## Improvement from data instead of better training method
>
> Thank you for this comment! We basically agree with the statement about *much of the performance boost appears to come from ..., rather than advancements in the model’s actual ability to learn distributions.* However, we respectfully clarify that our goal in this work is not to demonstrate that decoder-only autoregressive (AR) modeling has better learning ability than other learning methods. Instead, we aim to show that this modeling approach is at least competent in tasks like image generation, thereby enabling the research community to further explore its true advantage—an easy and elegant unification of various tasks.
>
> Specifically, we believe that the value of decoder-only AR models with discrete media representations lies not in outperforming specialized methods on individual tasks but in their potential to provide a simple, elegant, and extensible framework to unify various understanding and generation tasks across different modalities. This characteristic makes them a promising approach for achieving true unification. Given this innate strength, as long as their task-specific performance approaches that of specially designed methods, such models deserve a distinct place in the field. However, prior to our work, the performance gap between decoder-only AR models and mainstream methods in image generation is kind of too large, which largely hindered progress toward the ultimate goal. In contrast, our work is the first in the community to elevate the image generation performance of decoder-only models to a modern diffusion model level, establishing a critical foundation for their further development.

---

> > ### Comment · Reviewer_c5qN · 2024-11-24
> > **Thank you for the author responses**
> >
> > I have reviewed the author responses as well as the comments from other reviewers.
> >
> > Overall, many reviewers have raised concerns about the limited novelty and unclear positioning of the paper's contributions. While I echo this particular point, I think the work is acceptable overall, as I noted in my preliminary comments.
> >
> > Regarding the so-called "zero-shot" ability (which I believe needs to be adjusted and clarified further), I suspect that this phenomenon may not originate from this proposed Lumina-mGPT. Earlier works on image generation using discrete representations, if I remember correctly, such as VQ-GAN and VQ-Diffusion, have similar observations, where the reconstruction performance from the "generated" sequence tokens surpasses that of VQ-VAE. I am concerned that this may not provide significant insight into the decoder’s methodological design from this particular work. Also, larger models with prompting make it harder to diagnose this phenomenon precisely.
> >
> > At this stage, I remain borderline on this submission and am open to either decision, aligning with the consensus of the other reviewers.

---

> ### Author Response · Authors · 2024-11-23
> **Looking forward to your feedback**
>
> Dear Reviewer c5qN,
>
> Considering the discussion period is about to end, we sincerely look forward to your replies. We hope that we have addressed your concerns; if not, we are happy to listen to the follow-up questions. Thanks!
>
> Bests, Lumina-mGPT authors

---

> ### Author Response · Authors · 2024-11-24
> **Thank you so much for the response!**
>
> Dear Reviewer c5qN,
>
> Thank you very much for your thoughtful response. After carefully reading your feedback, we believe we have gained a deeper understanding of your concern. If we interpret your concern correctly, you are suggesting that the observed improvement in reconstruction quality should primarily be attributed to the inherent properties of VQ-VAE's discrete representation, and similar phenomena can be also observed in various generative models (e.g., autoregressive models, discrete diffusion models, etc.); in contrast, the improvement is not likely to stem from any particular advantage or uniqueness of (the components of) Lumina-mGPT—especially those claimed as contributions. Thus, we should avoid implying that this phenomenon constitutes an advantage or contribution specific to our work (Lumina-mGPT).
>
> **Upon reflection, we completely agree with this concern.**
>
> In fact, during writing the previous revision, our understanding was already basically aligned. We simply intended to emphasize that we have observed the phenomenon itself, rather than to claim a causal relationship between this phenomenon and any specific component or contribution of Lumina-mGPT. However, after reviewing your response, we re-examined our related statements and realized that our revisions were not thorough enough. **As a result, we have submitted a new revision with updates to Appendix B, and you may review it. The main changes include the following:**
> 1. We updated the section title from "Lumina-mGPT as VQ Codes Refiner" to "Discussions on Reconstruction Quality" to further avoid the aforementioned issue.
> 2. At the end of the section, we explicitly added the following statement: "It is worth noting that we believe this phenomenon should mainly be attributed to the intrinsic properties of VQ-VAE's discrete representation. We hypothesize that similar effects might also be observed in various generative models (e.g., autoregressive models, discrete diffusion models, etc.) and not due to any specific advantage or uniqueness of the decoder-only autoregressive model or other components of Lumina-mGPT, especially those claimed as contributions."
> 3. We changed several occurrences of the wording "Lumina-mGPT" to more general expressions, such as "Generative models such as Lumina-mGPT," to emphasize generality.
> 4. We moved this section from the main text to the appendix, as it is not directly related to the primary focus of the paper. (In fact, it is already done in the last revision).
>
> We would greatly appreciate your further feedback on these improvements.
>
> Once again, thank you for your time and efforts in helping us improve this work!
>
> Sincerely,
>
> The Lumina-mGPT Authors

---

### Official Review · Reviewer_UuZu · 2024-11-01

**Soundness:** 3
**Presentation:** 2
**Contribution:** 2
**Rating:** 5
**Confidence:** 3

**Summary:**

The paper proposes to finetune an autoregressive decoder-only model (initialized from Meta's Chameleon model) for image generation based on multimodal inputs. For this the paper initializes their model from the Chameleon checkpoint and finetunes the model for image generation and also adds additional multimodal tasks (e.g.,image editing, dense predictions like segmentation maps, spatial conditions, VQA, etc) to the training. The resulting model can generate photorealistic images in various aspect ratios and can also perform the multimodal tasks it was finetuned on.

**Strengths:**

The paper finetunes the Chamaleon model to add image generation and other multimodal tasks to the model. The original Chameleon model can also do this but the image generation model was not publicly released.
The paper also proposes a simple representation of the images to support various aspect ratios by adding information about height and width, as well as end-of-line tokens after each row of (latent) pixels.
The finetuning on multimodal tasks shows that the training approach also generalizes to tasks besides image generation.

**Weaknesses:**

There is limited novelty in the paper. Most of the work seems to come from the original Chameleon checkpoint with the image training being a relatively straight-forward finetuning approach.

The finetuning with multimodal tasks is also a relatively straight forward extension and other works have already shown that AR vision models can handle many different tasks and also can perform in-context learning (e.g., Sequential Modeling Enables Scalable Learning for
Large Vision Models, CVPR 2024).

Overall it's not clear to me what exactly is novel and different from Chameleon or other large VLMs.

**Questions:**

What exactly is different about Lumina-mGPT from other large VLMs? Is the training different? Is there some novelty in how the training is done?

---

> ### Author Response · Authors · 2024-11-20
> **Thank you for your comments!**
>
> Thank you for your comments and question!
>
> # How is Lumina-mGPT different from other works?
>
> First, from a broader perspective, most vision-language models that pivot around large language models (LLMs), often referred to as Multimodal LLMs (MLLMs), rely on pre-trained vision encoders like CLIP for visual understanding and pre-trained vision generators like Stable Diffusion for visual generation. While such approaches achieve good performance on specific tasks, they expose two key gaps: the gap between visual generation and understanding representations, and the gap between learning in the text and vision modalities.
>
> In contrast, **Lumina-mGPT** follows a different research trajectory. It adopts decoder-only autoregressive (AR) LLMs combined with discrete modality tokenization to unify all modalities into a shared discrete token space. The most significant advantage of this approach is its promising potential to provide an elegant, extensible, and unified framework for both understanding and generation across modalities, making it a compelling direction toward true multimodal unification. Against this backdrop, the differences between **Lumina-mGPT** and the two works mentioned—**Chameleon** and **Bai et al. (2024)[1]**—become evident:
>
> 1. **Chameleon** shares a similar goal to Lumina-mGPT. However, its value is limited by the absence of released image-generation capabilities. The defining strength of decoder-only AR models lies in their ability to unify both understanding and generation across modalities, but without generation capabilities, an understanding-only version cannot meet the broader expectations of such models. In contrast, Lumina-mGPT not only proposes but also open-sources its capabilities, providing the community with a foundation to further explore this domain. Moreover, based on the presented demos and reported quantitative metrics, Chameleon’s image-generation capabilities significantly lag behind those of Lumina-mGPT (Chameleon attains a score of 0.39 on the GenEval benchmark, while Lumina-mGPT attains 0.56).
>
> 2. **Bai et al. (2024)**, while undoubtedly an excellent work, focuses solely on the image modality without incorporating natural language. Natural language, however, is the most intuitive and user-friendly way for humans to interact with models. Its absence complicates tasks like generating content tailored to user needs, and often requires lengthy image prompts to specify target tasks. More importantly, language serves as the ideal "glue" for integrating diverse modalities within a shared context, which makes it an important part in fulfilling the primary advantage of decoder-only AR models in achieving broad unification.
>
> Furthermore, none of the existing works have demonstrated that decoder-only AR models can generate photo-realistic images at the level of modern diffusion models. Meanwhile, despite the simplicity of the proposed **UniRep**, it fills the blank in existing related works regarding variable aspect ratio image processing, and we believe that simplicity is usually not a bad thing in research.
>
> **For your information, our contributions are outlined as follows:**
>
> 1. We are the first (especially in the open-source domain) to demonstrate that a decoder-only AR model can achieve image generation performance comparable to modern diffusion models. Furthermore, by initializing the model with multi-modal generative pretraining, this capability can be achieved with relatively low computational resources (32 A100 GPUs * 7 days for 7B FP-SFT).
> 2. We propose UniRep, an image representation that empowers decoder-only AR models with the ability to flexibly generate images of varying aspect ratios.
> 3. Building on the strong image generation capabilities, we further explore Omni-SFT, an initial attempt to elevate the model into a unified, omni-potent model. Experimental results highlight the immense potential of decoder-only AR models in this direction.
> 4. We open-source the entire pipeline to encourage the community's further exploration of this topic.
>
> [1] Bai, Yutong, et al. "Sequential modeling enables scalable learning for large vision models." 2024.

---

> ### Author Response · Authors · 2024-11-23
> **Looking forward to your feedback**
>
> Dear Reviewer UuZu,
>
> Considering the discussion period is about to end, we sincerely look forward to your replies. We hope that we have addressed your concerns; if not, we are happy to listen to the follow-up questions. Thanks!
>
> Bests, Lumina-mGPT authors

---

> ### Author Response · Authors · 2024-11-30
> **Sincerely looking forward to your feedback**
>
> Dear Reviewer UuZu,
>
> As the extended discussion period is coming to an end, we would like to kindly remind you that we sincerely appreciate your further feedback. We hope that we have addressed your concerns regarding the clarity of novelty and the differences from existing works. We would also like to know if you have any other concerns or suggestions.
>
> Best regards,
> Lumina-mGPT Authors

---

### Official Review · Reviewer_uRrH · 2024-11-02

**Soundness:** 3
**Presentation:** 3
**Contribution:** 3
**Rating:** 6
**Confidence:** 4

**Summary:**

The paper "Lumina-mGPT" introduces a multimodal autoregressive model designed for flexible, photorealistic image generation from text, leveraging a decoder-only transformer and multimodal Generative PreTraining (mGPT). Lumina-mGPT unifies text and image processing within a single framework, using Flexible Progressive Supervised Finetuning (FP-SFT) for high-resolution, flexible-aspect image synthesis, and Omnipotent Supervised Finetuning (Omni-SFT) for a wide range of vision-language tasks, including segmentation and multiview generation. The model’s novel unambiguous image representation enhances its ability to generate variable-resolution images, and extensive multimodal pretraining enables impressive zero-shot capabilities across visual tasks.

**Strengths:**

The paper presents a systematic approach to fine-tuning through Flexible Progressive Supervised Fine-Tuning (FP-SFT) and Omnipotent Supervised Fine-Tuning (Omni-SFT). These methods are well-supported by empirical results, demonstrating that Lumina-mGPT generates high-quality, high-resolution images across various aspect ratios. Additionally, an extensive evaluation of zero-shot performance and attention visualizations offers insights into the model's generalizability and internal mechanisms.

The paper is well-organized and clearly articulates its goals, methodology, and findings. The authors effectively communicate complex technical concepts, ensuring that each design choice—such as the unambiguous image representation—is both motivated and illustrated, enhancing readability. While certain sections, such as the detailed explanations of supervised fine-tuning processes, may require technical background to fully appreciate, the overall clarity remains high.

Lumina-mGPT's primary contribution lies in bridging the gap between autoregressive (AR) models and diffusion-based image quality, with its open-source release further amplifying its impact.

**Weaknesses:**

The introduction and abstract emphasize the decoder-only architecture for text and image as a novel contribution, which is strange, given that several prior works—some even cited in this paper—have already employed similar architectures.

The effectiveness of Flexible Progressive Supervised Finetuning (FP-SFT) would benefit from more detailed analysis, especially around why certain parameter settings or resolution stages were chosen. Currently, the progressive finetuning approach appears somewhat arbitrary, and an ablation study exploring different finetuning configurations or showing specific benefits per stage would substantiate these choices.

**Questions:**

The Large World Model (LWM) paper introduces a multimodal autoregressive model capable of processing extensive sequences of video and book data, with a context length of millions of tokens. This enables it to perform tasks such as language understanding, image analysis, and video generation. In contrast, the Lumina-mGPT model is focused specifically on photorealistic text-to-image generation. Is this understanding accurate? Based on reading both papers, LWM appears to use a less effective VQ-VAE compared to Lumina-mGPT's VQ tokenizers; does this difference explain the variation in visual quality? Could the authors provide an analysis to identify the areas contributing the most to these performance gains?

While Lumina-mGPT demonstrates impressive performance across various tasks, additional examples in less conventional vision-language settings would help illustrate its adaptability and generalization. Could the authors provide results or comparisons on more challenging benchmarks or diverse vision-language tasks (e.g., object counting, spatial reasoning) that may better highlight Lumina-mGPT's strengths and weaknesses across the multimodal spectrum?

---

> ### Author Response · Authors · 2024-11-20
> **Thanks for your comments!**
>
> ## About Contribution
>
> Thank you very much for the comments, and we apologize for the lack of clarity in our initial submission. In the revision, we have clearly stated our contributions in the abstract and the introduction parts, and the relevant sections have been highlighted in red for easier review.
>
> Overall, our contributions are outlined as follows:
>
> 1. We are the first (especially in the open-source domain) to demonstrate that a decoder-only AR model can achieve image generation performance comparable to modern diffusion models. Furthermore, by initializing the model with multi-modal generative pretraining, this capability can be achieved with relatively low computational resources (32 A100 GPUs * 7 days for a 7B FP-SFT).
> 2. We propose UniRep, an image representation that empowers decoder-only AR models with the ability to flexibly generate images of varying aspect ratios.
> 3. Building on the strong image generation capabilities, we further explore Omni-SFT, an initial attempt to elevate the model into a unified, omni-potent model. Experimental results highlight the immense potential of decoder-only AR models in this direction.
> 4. We open-source the entire pipeline to encourage the community's further exploration of this topic.
>
>
> **Why claiming Decoder-only Architecture as a contribution**:
>
> First, we totally agree that the original expression was unclear, and we have carefully refined them in the revision. Meanwhile, we hope to kind note that in the initial submission, decoder-only architecture is considered as a **feature**, instead of **contribution**. Below is a more precise explanation of what we intended to convey:
>
> Compared to more complex designs like encoder-decoder architectures, decoder-only models have a significant innate advantage: they provide a simple, elegant, and extensible framework to unify various understanding and generation tasks across different modalities. This makes them a promising candidate for achieving true unification. However, the image generation capabilities of existing works in this paradigm—especially in the open-source community—remain limited, raising doubts about whether high-quality image generation is feasible for decoder-only models. This limitation casts a shadow over such models' potential as an all-encompassing solution. In contrast, our work is the first in the community to elevate the image generation performance of decoder-only models to a modern diffusion model level, establishing a critical foundation for their further development.
>
>   ## About the configuration of FP-SFT
>
> Thank you for the question! The progressive training strategy is a widely adopted approach in training diffusion models \[1\]\[2\]. Due to the quadratic complexity of transformers with respect to sequence length, directly training on high-resolution images results in low throughput and high resource costs. In contrast, a progressive strategy often achieves better visual quality at a fixed training cost.
>
> The specific resolution choices—512 → 768 → 1024—are based on various considerations. For instance, given that we consistently use 32 A100 GPUs for this project, lowering the resolution to 256 would decrease the computation-to-communication ratio and GPU utilization. Additionally, visualization revealed that the reconstruction quality at 256 resolution is relatively poor due to distortion introduced by VQVAE, so we chose 512 as the starting resolution. We selected 1024 as the final resolution because it is a standard in the field for high-resolution image generation. To bridge the gap between the starting and final resolutions, we introduced an intermediate stage at 768 resolution. Figure 4 in the manuscript illustrates the impact of each stage when using FP-SFT.
>
> Overall, we acknowledge that the choice of training configurations lacks rigorous validation. However, conducting extensive ablation studies is challenging for us due to the associated resource costs.
>
> [1] Zhuo, Le, et al. "Lumina-next: Making lumina-t2x stronger and faster with next-dit.", 2024
>
> [2] PKU-Yuan Lab and Tuzhan AI etc. "Open-Sora-Plan", 2024

---

> > ### Author Response · Authors · 2024-11-21
> > **Response to the raised questions**
> >
> > ## Comparison with LWM[1]
> >
> > Thank you for bringing up this fascinating question! From a modeling perspective, both LWM and our work fall under the category of decoder-only AR models. However, there are significant differences in their focus:
> >
> > LWM primarily emphasizes showcasing ring attention as an effective technique for extending the context window, demonstrating its strong capabilities and promising results in building multi-modal, long-context LLMs. While it incorporates image and video modalities, its main focus is on the feasibility of training a model with such an extended context per se, rather than prioritizing the quality of generation. Consequently, the datasets used in LWM for both image and video are large-scale but relatively low-quality (in terms of aesthetics, textual description quality, and image-text alignment). Furthermore, LWM does not adopt strategies like resolution-level progressive training to improve overall generation capabilities and training efficiency. As we noted in Appendix C, the quality of the VQVAE also significantly impacts image generation quality. However, this factor may be less significant compared to other aforementioned issues. Lastly, it is worth noting that LWM utilized computational resources equivalent to 450 A100 GPUs, while the FP-SFT of the Lumina-mGPT 7B model required only 32 A100 GPUs for 7 days, highlighting that these two works fall into different categories.
> >
> > [1] Liu, Hao, et al. "World model on million-length video and language with ringattention." 2024.
> >
> >
> > ## More evaluations
> >
> > During this rebuttal, we have additionally evaluated the text-to-image ability of Lumina-mGPT on popular benchmarks to better reflect its position in the field, and the results are as follows:
> >
> > |             | T2I-CompBench |        |         | GenEval | DPG-Bench |
> > | ----------- | ------------- | ------ | ------- | ------- | --------- |
> > |             | Color         | Shape  | Texture | Overall | Average   |
> > | Lumina-Next | 0.5088        | 0.3386 | 0.4239  | 0.46    | 75.66     |
> > | SDv2.1      | 0.5694        | 0.4495 | 0.4982  | 0.50    | -         |
> > | SDXL        | 0.6369        | 0.5408 | 0.5637  | 0.55    | 74.65     |
> > | Chameleon   | -             | -      | -       | 0.39    | -         |
> > | Lumina-mGPT | 0.6371        | 0.4727 | 0.6034  | 0.56    | 79.68     |
> >
> > As shown in the table, Lumina-mGPT demonstrates a significant improvement over Chameleon on the GenEval benchmark. Furthermore, Lumina-mGPT outperforms Lumina-Next, a modern diffusion transformer trained using the same text-to-image data as Lumina-mGPT, and we infer that generative pre-training may be the reason behind the performance gap. Additionally, Lumina-mGPT shows competitive performance compared to SDXL, while maintaining a stable advantage over SDv2.1, providing an intuitive picture of Lumina-mGPT’s position in the field.

---

> ### Author Response · Authors · 2024-11-23
> **Looking forward to your feedback**
>
> Dear Reviewer uRrH,
>
> Considering the discussion period is about to end, we sincerely look forward to your replies. We hope that we have addressed your concerns; if not, we are happy to listen to the follow-up questions. Thanks!
>
> Bests, Lumina-mGPT authors

---

> ### Author Response · Authors · 2024-11-30
> **Sincerely looking forward to your feedback**
>
> Dear Reviewer uRrH,
>
> As the extended discussion period is coming to an end, we would like to kindly remind you that we sincerely appreciate your further feedback. We hope that we have addressed your concerns and questions. We would also like to know if you have any other concerns or suggestions.
>
> Best regards, Lumina-mGPT Authors

---

### Official Review · Reviewer_AimJ · 2024-11-02

**Soundness:** 3
**Presentation:** 4
**Contribution:** 3
**Rating:** 8
**Confidence:** 4

**Summary:**

This paper introduces Lumina-mGPT, a multimodal autoregressive model that excels in generating photorealistic images from textual descriptions, as well as performing a variety of other vision and language tasks. Built on a pre-trained decoder-only transformer architecture, the model adeptly processes multimodal token sequences. Key innovations of Lumina-mGPT include Effective Multimodal Generative Pretraining (mGPT) using extensive interleaved text-image datasets, Flexible Progressive Supervised Fine-tuning (FP-SFT), and Omnipotent Supervised Finetuning (Omni-SFT). These developments enable the model to produce images of varying resolutions and accommodate a broad spectrum of vision-language tasks, marking a notable advancement in flexibility and task integration over traditional autoregressive approaches.

**Strengths:**

+ Omni-SFT within Lumina-mGPT effectively unifies a diverse array of tasks within a single model framework, demonstrating its extensive multitasking capabilities across various complex applications such as text-to-image synthesis, image captioning, image editing, spatial-conditional image generation, and more.
+ The use of mGPT for initial training, followed by sophisticated fine-tuning strategies like FP-SFT and Omni-SFT, establishes a solid foundation for generating high-quality, photorealistic images while adeptly handling a wide range of multimodal tasks.
+ Lumina-mGPT exhibits exceptional flexibility in generating images across different resolutions and aspect ratios—a significant advantage over many existing models.
+ By effectively bridging the gap between autoregressive and diffusion methods, Lumina-mGPT achieves remarkable visual aesthetics and detailed image rendering without the need for cascading models,
+ The paper ingeniously integrates various advanced concepts from the literature and existing models to enhance the training and inference capabilities of Lumina-mGPT. This includes adopting classifier guidance from diffusion models and stabilization techniques used in large language models, which collectively contribute to the robustness and efficiency of the framework.

**Weaknesses:**

- The paper falls short in providing comprehensive comparative metrics, such as FID scores, with only limited comparisons featured in Table 2 against the Chameleon model. A more extensive range of benchmarking against current state-of-the-art (SoTA) models is essential to objectively evaluate the model’s performance and its advancements over existing methodologies. Including a broader set of metrics would significantly clarify the model's positioning and contribution within the broader field.
- Although the paper showcases zero-shot capabilities in enhancing visual details, it only offers qualitative results. Incorporating quantitative evaluations would provide a more robust comparison against other models and substantiate the claimed improvements.
- While Omni-SFT demonstrates the model's capability to handle diverse tasks, the absence of quantitative results in a controlled evaluation test limits the understanding of its performance, especially in comparison to specialized models. Providing such data would help gauge the effectiveness of Omni-SFT and its relative performance across different tasks.
- The paper highlights the performance benefits of Omni-SFT fine-tuning on top of the Lumina model but lacks a comparative analysis with scenarios where Omni-SFT is fine-tuned starting from the initial mGPT. A direct comparison would help validate the advantage of transitioning from Lumina to Omni for task unification, ensuring that this methodological choice yields tangible benefits.

**Questions:**

- In the appendix, you observe that *"with the CFG increasing, the quality of generated images improves, proving the effectiveness of the classifier-free guidance in this context."*. This finding appears to contrast with typical outcomes in diffusion models, where increasing the CFG beyond a certain threshold often results in diminished image quality. Could you elucidate the reasons behind this differing impact of CFG in your model compared to traditional diffusion models?

---

> ### Author Response · Authors · 2024-11-20
> **Thank you for your recognition of our work!**
>
> ## About quantitative evaluation
>
> Thank you very much for your valuable feedback! We fully agree that adding numerical metrics would help clarify the performance and positioning of Lumina-mGPT. In response, we have tested Lumina-mGPT on popular T2I benchmarks, and the results have been added to the revision (Sec. 3.1). For your convenience, we also show the results here:
>
> |             | T2I-CompBench |        |         | GenEval | DPG-Bench |
> | ----------- | ------------- | ------ | ------- | ------- | --------- |
> |             | Color         | Shape  | Texture | Overall | Average   |
> | Lumina-Next | 0.5088        | 0.3386 | 0.4239  | 0.46    | 75.66     |
> | SDv2.1      | 0.5694        | 0.4495 | 0.4982  | 0.50    | -         |
> | SDXL        | 0.6369        | 0.5408 | 0.5637  | 0.55    | 74.65     |
> | Chameleon   | -             | -      | -       | 0.39    | -         |
> | Lumina-mGPT | 0.6371        | 0.4727 | 0.6034  | 0.56    | 79.68     |
>
> As shown in the table, Lumina-mGPT demonstrates a significant improvement over Chameleon on the GenEval benchmark. Furthermore, Lumina-mGPT outperforms Lumina-Next, a modern diffusion transformer trained using the same text-to-image data as Lumina-mGPT, and we infer that generative pre-training may be the reason behind the performance gap. Additionally, Lumina-mGPT shows competitive performance compared to SDXL, while maintaining a stable advantage over SDv2.1, providing an intuitive picture of Lumina-mGPT’s position in the field.
>
> ## About the zero-shot capability in enhancing visual details
>
> As also pointed out by reviewer c5qN, we believe that our previous statement of *zero-shot* is inappropriate and we have changed the statement from *a zero-shot ability* to *an interesting observation*, please see Appendix.B for details. In fact, the *zero-shot visual quality enhancement* in the initial submission refers to a phenomenon we observed during our experiments:
>
>  given two data flows:
>
> 1. Image $\xrightarrow{\text{VQVAE Encoder}}$ latent$\xrightarrow{\text{VQVAE Decoder}}$ Reconstruction1
> 2. Image $\xrightarrow{\text{VQVAE Encoder}}$ latent$\xrightarrow{\text{Lumina-mGPT using editing system prompt with instruction "no-edit"}}$ latent2$\xrightarrow{\text{VQVAE Decoder}}$ Reconstruction2
>
> We surprisingly find that the quality of Reconstruction2 sometimes surpasses that of Reconstruction1. Note that, while we train on image editing tasks, the model is not trained on the "no-edit" instruction, which led us to originally use the term "zero-shot". This phenomenon is counter-intuitive, as Lumina-mGPT only has access to VQVAE latents during training and cannot access original image, so intuitively the quality of VQVAE reconstruction should somehow build an upper bound for Lumina-mGPT's image generation quality, which contradicts our observations. We find this observation intriguing and believe it may provide some insights (e.g. it may indicate that the latents encoded by VQVAE encoder could contain noises of certain pattern that can be learned and even correctly by Lumima-mGPT after training), so we added the contents to the manuscript. However, since the model is not explicitly trained on the "no-edit" instruction, the process is difficult to control, and quantitative evaluation in a controlled setting is challenging. Besides, our purpose is to present this intriguing observation, rather than claim that Lumina-mGPT was already a good vq code refiner. in the revision, we have carefully refined the expressions for clarity.
>
> ## About quantitative results of Omni-SFT
>
> Thank you very much for your comment. We fully understand that the true advantage of decoder-only AR models lies in their ability to unify different tasks and modalities. We also agree that performance across a wide range of omnipotent tasks is crucial for evaluating such models. However, considering the current progress in this field—the quality of image generation, which is a prerequisite for achieving omni-potency, is still very limited—in this work we have focused primarily on improving image generation quality, as well as (building on this improved quality) on conducting a **preliminary** exploration of omni-potency. We are excited to see the initial ability of the model to address various tasks after Omni-SFT. However, we must acknowledge that the current capabilities are still far from being on a level that allows for a reasonable comparison with task-specific models. We look forward to further exploring this topic in future work.
>
> ## About initializing Omni-SFT from initial mGPT
>
> We would kindly note that the mGPT we use, namely the publicly-released Chameleon model, **does not** have image generation capabilities. Therefore, we believe that it is quite reasonable and straightforward to first build the image generation ability (with flexible height-width ratio) through FP-SFT, and subsequently apply Omni-SFT to learn tasks that are essentially framed as conditional image generation problems.

---

> ### Author Response · Authors · 2024-11-21
> **About the trend of image quality variation with respect to CFG**
>
> ## About the trend of image quality variation with respect to CFG
>
> Thank you very much for pointing this out. We sincerely apologize for the lack of rigor in the initial submission. In Fig.7 of the original submission, the image quality indeed shows an improving trend as the CFG value increases; however, this conclusion is limited within a small interval (1.0 to 8.0). Our previous description may have mistakenly implied that this trend holds universally across the entire domain. In the revision, see Appendix.E and Fig.8 in the revision, we have included test results over a broader range (1-400), which demonstrate that, overall, the image generation quality follows a trend similar to diffusion models: it initially improves as CFG increases but then deteriorates. We have also revised the relevant phrasing to eliminate any ambiguity.

---

> > ### Comment · Reviewer_AimJ · 2024-11-27
> >
> > Thank you for your responses and for providing the evaluation tables for compositionality benchmarks (T2I-CompBench), as well as GenEval and DPG-Bench. I appreciate the effort in addressing the concerns and providing additional insights.
> >
> > Regarding T2I-CompBench, while the model demonstrates strong performance, it still lags behind some of the more recent state-of-the-art models like SD 3, which, for example, achieves around 80% on the color dataset. That said, your model still exhibits commendable results and represents a valuable contribution to the field.
> >
> > I do believe, however, that a more thorough and comprehensive quantitative evaluation would further strengthen the paper. Despite this, I find the paper to be well-written, technically sound, and a valuable contribution, and I will maintain my rating.

---

> > > ### Author Response · Authors · 2024-11-29
> > >
> > > Thank you for recognizing the contributions of our work! We greatly appreciate your insightful comments, which have been invaluable in improving our work. We will continue improving Lumina-mGPT, particularly in terms of data, to fully showcase its full potential to compete with sota diffusion-based methods.

---

> ### Author Response · Authors · 2024-11-23
> **Looking forward to your feedback**
>
> Dear Reviewer AimJ,
>
> Considering the discussion period is about to end, we sincerely look forward to your replies. We hope that we have addressed your concerns; if not, we are happy to listen to the follow-up questions. Thanks!
>
> Bests, Lumina-mGPT authors

---

### Official Review · Reviewer_nbPV · 2024-11-04

**Soundness:** 2
**Presentation:** 3
**Contribution:** 2
**Rating:** 6
**Confidence:** 4

**Summary:**

This work presents Lumina-mGPT, a series of autoregressive models for multimodal understanding and generation. With a pre-trained decoder-only transformer, Lumina-mGPT allows for a unified framework for multi-model modeling. The work shows that multimodal generative pretraining is the key towards general multimodal capabilities. Furthermore, flexible progressive supervised fine-tuning, as proposed by the work, allows high-aesthetic image generation at flexible resolutions. Finally, omnipotent supervised fine-tuning enables Lumina-mGPT to achieve task unification in visual generation and understanding.

**Strengths:**

* The proposed Unambiguous image Representation, when used with Flexible Progressive Supervised Finetuning (FP-SFT), allows the method to generate images of varying resolutions.
* While prior method Chameleon can only perform vision-language and text-only tasks, Lumina-mGPT achieves visual recognition tasks (e.g., segmentation, depth prediction) and controllable generation as well as image editing, which makes Lumina-mGPT a unified model for various downstream applications.
* The performance on VQA benchmarks significantly improves over baseline Chameleon, demonstrating the effectiveness of the proposed model.

**Weaknesses:**

* The contributions of this work is not clearly described. While the work claims mGPT to be a key insight, the importance of mGPT is not discovered by the work, as the mGPT model is adapted from Chameleon. The difference is that Lumina-mGPT performs fine-tuning on Chameleon.
* The proposed Unambiguous image Representation (Uni-Rep) is only applied at the supervised fine-tuning stage. This creates gaps between the image representation in pre-training and fine-tuning.
* The method claims that baseline Chameleon shows degraded visual quality compared to diffusion methods (L242-244). However, this work does not perform quantitative evaluations on the quality of the generated images to illustrate whether the method is able to overcome this limitation. Metrics such as FID and/or results on benchmarks such as T2I-CompBench [1] are needed to show the performance improvements against other related works.
* No comparisons are provided with text-only LLMs on text-only benchmarks such as MMLU. The example dialogues are insufficient in evaluating the model's capabilities in text-only tasks.
* No human evaluation is performed to assess the quality of multi-modal generation.

[1] T2I-CompBench: A Comprehensive Benchmark for Open-world Compositional Text-to-image Generation. K. Huang, et al. https://arxiv.org/abs/2307.06350

**Questions:**

The comparisons and evaluations are insufficient for the work. Specifically, a few questions are still to be addressed:
* How does Lumina-mGPT compare with diffusion-based methods in terms of image generation?
* How does Lumina-mGPT perform on text-only tasks, comparing with other text-only LLMs?

---

> ### Author Response · Authors · 2024-11-20
> **Thanks for your comments!**
>
> ## About contribution
>
> Thank you very much for the comment. We acknowledge that our initial submission lacked a clear enumeration of our contributions. Following your comment, in the revision, we have clarified our contributions in the abstract and the introduction parts. For your convenience, the outlined contributions are as follows:
>
> 1. We are the first (especially in the open-source domain) to demonstrate that a decoder-only AR model can achieve image generation performance comparable to modern diffusion models. Furthermore, by initializing the model with multi-modal generative pretraining, this capability can be achieved with relatively low computational resources (32 A100 GPUs * 7 days for 7B FP-SFT).
> 2. We propose UniRep, an image representation that empowers decoder-only AR models with the ability to flexibly generate images of varying aspect ratios.
> 3. Building on the strong image generation capabilities, we further explore Omni-SFT, an initial attempt to elevate the model into a unified generalist. Experimental results underscore the promising potential of this direction.
> 4. We open-source the entire pipeline to encourage the community's further exploration of this topic.
>
> We also agree that, though multi-modal generative pre-training is an important design choice, claiming it as the core insight is improper, and we have also modified relevant statements in the revision.
>
> ## About the gap introduced by UniRep
>
> Thank you very much for your comment on the gap introduced by UniRep. While we acknowledge the gap mentioned does exist to some extent, we sincerely believe that this may not significantly affect our model for the following reasons:
>
> 1. First, please kindly note that the image generation capability of Chameleon has not been released yet, so we are currently initializing from a model *without any image generation capability*. In this context,  the gap introduced by UniRep is relatively small when compared to the much larger gap between a model with no image generation ability and one that can generate images.
>
> 2. As mentioned in Sec.2.2.2 of the manuscript, UniRep is indeed a must instead of a decoration if we are to support the model with variable height/width ratios. This makes the gap something that we cannot avoid. We fully agree that it makes more sense to apply UniRep also to the pre-training stage. However, pre-training comes with enormous resource costs (e.g., Chameleon 7B used 856,481 GPU hours), which are unfortunately beyond the reach of most research labs.
>
> 3. On the other hand, despite the existence of this gap, there is no evidence suggesting that it significantly impacts the generation quality. Both the qualitative examples provided in our initial submission, as well as the quantitative results presented in this rebuttal (shown below), demonstrate the superior image generation quality of Lumina-mGPT.
>
> ## About evaluations on image generation
>
> Thank you so much for this valuable feedback! We sincerely agree with your opinion that quantitative results are necessary to support the following claims:
>
> 1. Chameleon demonstrates degraded image generation quality, while Lumina-mGPT performs significantly better.
> 2. Lumina-mGPT exhibits competitive capabilities relative to diffusion models.
>
> We have since conducted evaluations on popular text-to-image benchmarks, and the results are as follows:
>
> |             | T2I-CompBench |        |         | GenEval | DPG-Bench |
> | ----------- | ------------- | ------ | ------- | ------- | --------- |
> |             | Color         | Shape  | Texture | Overall | Average   |
> | Lumina-Next | 0.5088        | 0.3386 | 0.4239  | 0.46    | 75.66     |
> | SDv2.1      | 0.5694        | 0.4495 | 0.4982  | 0.50    | -         |
> | SDXL        | 0.6369        | 0.5408 | 0.5637  | 0.55    | 74.65     |
> | Chameleon   | -             | -      | -       | 0.39    | -         |
> | Lumina-mGPT | 0.6371        | 0.4727 | 0.6034  | 0.56    | 79.68     |
>
> As shown, Lumina-mGPT demonstrates a significant improvement over Chameleon on the GenEval benchmark. Furthermore, Lumina-mGPT outperforms Lumina-Next, a modern diffusion transformer trained using the same text-to-image data as Lumina-mGPT, and we guess that generative pre-training may be the reason behind this performance gap. Additionally, Lumina-mGPT shows competitive performance compared to SDXL, while maintaining a stable advantage over SDv2.1, providing an intuitive picture of Lumina-mGPT’s position in the field.
>
> The results have also been added to the revision (Sec.3.1).
>
> ## About text-only capability
> We fully acknowledge the importance of text-only capabilities. However, our work primarily focuses on the model's performance in image-text-related tasks, particularly in image generation. The text-only performance depends on factors such as the quality of the pretrained model and the pure text data used in supervised fine-tuning (SFT), which are orthogonal to the focus of this work.

---

> ### Author Response · Authors · 2024-11-23
> **Looking forward to your feedback**
>
> Dear Reviewer nbPV,
>
> Considering the discussion period is about to end, we sincerely look forward to your replies. We hope that we have addressed your concerns; if not, we are happy to listen to the follow-up questions. Thanks!
>
> Bests, Lumina-mGPT authors

---

> ### Author Response · Authors · 2024-11-30
> **Sincerely looking forward to your feedback**
>
> Dear Reviewer nbPV,
>
> As the extended discussion period is coming to an end, we would like to kindly remind you that we sincerely appreciate your further feedback. We hope that we have addressed your concerns regarding evaluation, the clarity of novelty, etc. We would also like to know if you have any other concerns or suggestions.
>
> Best regards,
> Lumina-mGPT Authors

---

> > ### Comment · Reviewer_nbPV · 2024-12-01
> >
> > The reviewer would like to thank the authors for the clarifications. The clarifications addressed several of the reviewer's concerns.
> >
> > 1. As for the contribution part, the reviewer acknowledges the performance gains compared to Chameleon and several previous baseline diffusion models.
> > 2. The argument that "the image generation capability of Chameleon has not been released yet, so we are currently initializing from a model without any image generation capability" and "the gap introduced by UniRep is relatively small when compared to the much larger gap between a model with no image generation ability and one that can generate images" is not valid. Although the image decoder has not been released, the sequence model is trained with fixed image resolution in pre-training. If we train an image decoder with the released encoder, there would technically be no gap between the formulation of model pre-training and the actual use at inference time. In contrast, UniRep introduces a gap between pre-training and fine-tuning as well as inference. The performance gain, as argued by the authors, is affected by several factors, including but not limited to UniRep. The reviewer would like to see ablations for the model trained with/without UniRep to see if the gap degrades the performance for fixed resolution generation. However, the reviewer acknowledges UniRep's flexibility in generating images of varying aspect ratios.
> > 3. The additional results on several benchmarks are promising.
> >
> > The reviewer would therefore like to raise the score.

---

> > > ### Author Response · Authors · 2024-12-02
> > > **Thank you for the response!!**
> > >
> > > Dear Reviewer nbPV,
> > >
> > > Thank you very much for your response and for raising the score. We are glad to see that your concerns have been addressed. We also greatly appreciate you pointing out imprecise expressions in our rebuttal.
> > >
> > > Once again, thank you for the time and effort you have put into reviewing our submission.
> > >
> > > Sincerely,
> > > Lumina-mGPT Authors

---

### Meta-Review · Area_Chair_XtZQ · 2024-12-21

**Metareview:**

This paper receive the ratings of 6, 6, 5, 5, 8, where the reviewers provided mixed feedback. In this paper, the authors introduce Lumina-mGPT, a decoder-only autoregressive multimodal model, designed to unify vision-language tasks while enabling flexible, photorealistic image generation. The paper emphasizes interesting techniques, such as Flexible Progressive Supervised Finetuning (FP-SFT) and Omnipotent Supervised Finetuning (Omni-SFT), for achieving scalability and task unification.

Strengths:
- The problem addressed by Lumina-mGPT is timely and relevant, with potential applications across multimodal AI.
- The proposed FP-SFT and Omni-SFT techniques are interesting attempts at addressing scalability and unification.
- The paper provides both qualitative and quantitative evaluations, with promising results on photorealistic image generation.

Area for improvements:
- Several reviewers pointed out that the novelty of this work is limited, especially as the differences between Lumina-mGPT and its predecessor, Chameleon, are minimal.
- Reviewers found the evaluation lacking in benchmarking against state-of-the-art diffusion models, particularly under identical training setups. This weakens the claim of performance parity with modern diffusion-based systems.
- The reliance on pretrained components (e.g., Chameleon) without detailed insights into training data and methodologies limits the transparency and reproducibility of the work.
- Ineffective instruction following, inconsistent artifact handling, and limited multilingual capabilities, further undermine the model's claimed generalizability.
- Clarity: the manuscript seems to overclaim certain achievements, which are not convincingly demonstrated.

While the work explores an important direction with some promising ideas, the current status does not meet the high standards of empirical rigor, clarity, and reproducibility required for acceptance at ICLR. We encourage the authors to address the area of improvements and resubmit to a future venue after further refinement and evaluation.

**Additional Comments On Reviewer Discussion:**

The authors’ rebuttal clarified some technical aspects but did not sufficiently address the primary concerns, especially related to the limited novelty, raised by the reviewers. The overall discussions were smooth and productive, allowing the authors to better articulate their contributions (despite one reviewer not very actively engaging in discussions during the rebuttal period)

The reviewers provided varied assessments of the paper, highlighting several areas for improvement. While the authors addressed these concerns during the rebuttal, the responses did not fully resolve the reviewers' key reservations. One recurring concern was that the proposed approach lacked sufficient distinction from existing methods, particularly due to its reliance on established architectures and datasets. Although the rebuttal emphasized how the work advances the state of the art, the explanations did not fully convince the reviewers of its originality or significance.

The reviewers also noted that the empirical evaluations could be strengthened, particularly through more comprehensive comparisons with strong baselines. Additionally, questions were raised regarding the broader impact and generality of the contributions. While the rebuttal outlined potential applications, it would benefit from more robust empirical evidence to substantiate these claims and demonstrate the model's practical utility and versatility.

---

### Decision · Program_Chairs · 2025-01-22

Reject